# Bayesian Concept Bottleneck Models with LLM Priors

**Jean Feng, Avni Kothari, Luke Zier**
University of California,
San Francisco

**Chandan Singh**
Microsoft Research

**Yan Shuo Tan**
National University
of Singapore

## Abstract

Concept Bottleneck Models (CBMs) have been proposed as a compromise between white-box and black-box models, aiming to achieve interpretability without sacrificing accuracy. The standard training procedure for CBMs is to predefine a candidate set of human-interpretable concepts, extract their values from the training data, and identify a sparse subset as inputs to a transparent prediction model. However, such approaches are often hampered by the tradeoff between exploring a sufficiently large set of concepts versus controlling the cost of obtaining concept extractions, resulting in a large interpretability-accuracy tradeoff. This work investigates a novel approach that sidesteps these challenges: BC-LLM iteratively searches over a potentially infinite set of concepts within a Bayesian framework, in which Large Language Models (LLMs) serve as both a concept extraction mechanism and prior. Even though LLMs can be miscalibrated and hallucinate, we prove that BC-LLM can provide rigorous statistical inference and uncertainty quantification. Across image, text, and tabular datasets, BC-LLM outperforms interpretable baselines and even black-box models in certain settings, converges more rapidly towards relevant concepts, and is more robust to out-of-distribution samples. [1]

## 1 Introduction

Although machine learning (ML) algorithms have demonstrated remarkable predictive performance, many lack the interpretability and transparency necessary for human experts to audit their accuracy, fairness, and safety [1, 2]. This has limited their adoption in high-stakes applications such as medicine [3] and settings where regulatory agencies require algorithms to be explainable [4]. Recent works have explored Concept Bottleneck Models (CBMs) [5, 6] as a potential solution: these methods leverage black-box algorithms to extract a small number of interpretable *concepts*, which are subsequently processed by a fully transparent tabular model to predict a target label. Thus, in principle, CBMs are much safer for use in high-stakes applications because humans can audit and, if necessary, modify the extracted concepts to fix predictions.

Nevertheless, CBMs have yet to fully realize this promise. Early CBM methods were difficult to scale because they relied on human experts to specify and annotate concepts on training data. Recent works have proposed using LLMs to suggest and even annotate concepts [7–10], as LLMs are cheaper, often have sufficient world knowledge to hypothesize useful concepts, and can provide (relatively) high-quality concept annotations [9, 10]. This introduces new problems such as LLM hallucinations and inconsistencies but, more critically, does not resolve the fundamental problem that standard CBMs sacrifice a significant amount of accuracy to gain interpretability [11]. Recent proposals try to address this by encoding concepts using "soft" continuous values, such as through embedding similarity [8], rather than "hard" binary values. However, "soft" CBMs can leak information about the label that would otherwise not have been available [12–16]; this risk is particularly severe when

---

[1] Code for running BC-LLM and reproducing results in the paper are available at `https://github.com/jjfeng/bc-llm`.

39th Conference on Neural Information Processing Systems (NeurIPS 2025).

concept extraction is jointly trained with the final label predictor [15].[2] Because the definitions of soft concepts are ambiguous, humans can no longer easily interpret, audit, or intervene on such CBMs.

The aim of this work is to minimize the accuracy gap without sacrificing the interpretability of the extracted concepts. To this end, we revisit a known pain point of CBMs: Current training procedures require a pool of candidate concepts to be identified *a priori*, but ensuring this pool contains all relevant concepts is difficult, particularly in domains that are yet to be scientifically well-understood and where there are an *infinite* number of potentially relevant concepts [17, 10]. For instance, for the task of predicting hospital readmission, there are an infinite number of factors affecting a patient's risk. Even a single concept like "smoking status" has an infinite number of refinements, such as the correlated concept of "whether a patient has quit smoking," the broader concept of "substance use," and more. As shown in prior work, when the candidate pool misses important concepts, the resulting CBMs may be inaccurate and even mislead users as to which concepts are truly relevant [18].

Rather than specifying concepts upfront, we investigate *iterative* refinement of concepts in a CBM with the assistance of an LLM. This updates a classical modeling paradigm for the LLM era: prediction models were traditionally designed through the collaboration between domain experts and data scientists, in which they iteratively refine/engineer features for simple tabular models. LLMs can help significantly accelerate this co-design process and scale up the number of iterations, but they are also imperfect query engines that can hallucinate, suggest bad candidate concepts, incorrectly annotate concepts, and may not even be self-consistent in their prior beliefs [19]. To set up guardrails, we "wrap" the LLM within a Bayesian posterior inference procedure to explore concepts with the help of an LLM *in a statistically principled manner*. This approach, which we refer to as **B**ayesian **C**oncept bottleneck models with **LLM** priors (BC-LLM), offers the following benefits:

- BC-LLM finds relevant concepts, even in settings where there is little to no prior knowledge: Unlike prior works that soley rely on a prespecified set of candidate concepts, BC-LLM explores a potentially infinite set of concepts data-adaptively and converge towards the true ones.
- BC-LLM substantially improves the interpretability-accuracy tradeoff in real-world datasets: Experiments across multiple datasets and modalities (text, images, and tabular data) show that BC-LLM selects human-interpretable concepts while outperforming comparator methods, often even black-box models. Moreover, when assisting a hospital's data science team to revise an existing tabular ML model, clinicians found BC-LLM to be substantially more interpretable and actionable.
- BC-LLM corrects LLM mistakes in a statistically rigorous manner to provide calibrated uncertainty quantification: Mistakes in the LLM's prior are fully allowed and even expected, as they are corrected in the Metropolis accept/reject step. Moreover, BC-LLM provides uncertainty quantification of the selected concepts, which improves performance in both in-distribution and out-of-distribution (OOD) settings. We prove that BC-LLM converges to the correct concepts, asymptotically, even when the LLM prior is poor or fails to be self-consistent. The Bayesian framework also naturally protects against overfitting and overconfidence in selected concepts.
- BC-LLM is cost-effective: BC-LLM selectively annotates $O(KT)$ concepts, where $K$ is the number of concepts in the CBM and $T$ is the number of outer loops; we often use only $K \approx 5$ and $T \approx 5$. In contrast, existing methods annotate *hundreds or thousands* of prespecified concepts [8].

## 2 Related work

Prior CBM methods train on observations with hand-annotated concepts [5, 6, 20] and, more recently, concepts annotated by an LLM or vision language model (VLM) [21, 9, 10]. These methods are known to be highly sensitive to the selected set of concepts [18]. More recent works have suggested using LLMs to prespecify the list of candidate concepts instead [7, 8, 22–26], but this requires the LLM to have accurate prior knowledge about the supervised learning task. We note that many of these works also make use of "soft concepts" and therefore suffer from information leakage [12–16].

To find relevant concepts in a more data-driven manner, recent works have proposed taking an iterative approach [17, 27–29]. These methods employ a boosting-like approach, where the LLM is provided a set of misclassified examples and asked to add/revise concepts to improve performance. However,

---

[2]For instance, a human would fill in 0 when asked "does this bird have a gray breast" if there is no gray breast, but a "soft" CBMs may extract 0.2 if it sees a blue breast or gray head. This provides additional hints for downstream prediction.

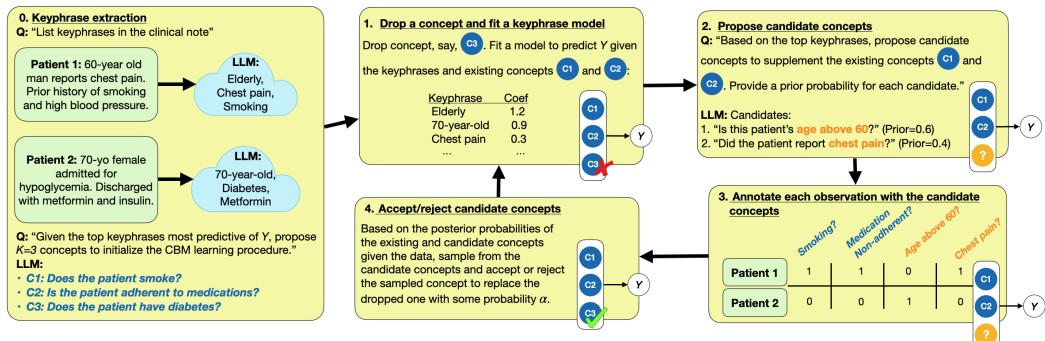

Figure 1: BC-LLM is initialized by having the LLM hypothesize the top concepts based on keyphrases extracted from each observation (Step 0). The concepts are then iteratively refined by dropping a concept (Step 1), querying the LLM for candidate replacements (Step 2), annotating each observation with the candidate concepts using the LLM (Step 3), and determining which, if any, of the candidate concepts to accept (Step 4).

the LLM often struggles to identify relevant patterns from the presented examples, so the resulting concept proposals are far from optimal.

An alternative strategy is to conduct post-hoc distillation of a black-box model into interpretable concepts [30–34], which can be used to learn an interpretable prediction model. Nevertheless, prior work has shown that the generated explanations are often not faithful to the black-box model, leading to poorer performance from the interpretable model [35].

Another related line of work uses LLMs to describe conceptual differences between pairs of datasets [36–38] or to directly describe a black-box function using natural-language concepts [39–42]. However, existing methods are primarily suited for simple classification tasks, as they essentially learn a CBM with a single concept rather than a set of concepts.

Finally, recent works have considered combining LLMs with Bayesian techniques [43–45]. Some have suggested that In-Context Learning (ICL) can be viewed as LLMs conducting Bayesian inference [46, 47], though recent work has proven that ICL is not fully consistent with Bayesian inference [48]. In contrast, this work studies how we can iteratively discover new concepts to include using LLMs using a Bayesian inference procedure, accounting for imperfections in LLM reasoning.

## 3   Method: BC-LLM

Consider a dataset $\mathcal{D} = \{(x_1, y_1), \ldots, (x_n, y_n)\}$, where $x_i$ are model inputs and $y_i$ are labels. Let $\mathbf{X} := (x_1, x_2, \ldots, x_n)$ and $\mathbf{y} := (y_1, \ldots, y_n)$. A *concept* $c$ defines a binary- or real-valued function $\phi_c$ of a model input, e.g. "Does the doctor note describe an elderly patient?" where 1=yes and 0=no. A CBM selects a fixed number of $K$ concepts $\mathbf{c} = (c_1, \ldots, c_K)$ and fits a parametric model for $y$ using the extractions $(\phi_{c_1}(x), \ldots, \phi_{c_K}(x))$. For simplicity of presentation, our discussion will focus on logistic regression (LR)-based CBMs, although other choices are possible. In this case, the data is modeled by $p\left(Y = 1 | X = x, \boldsymbol{\theta}, \mathbf{c}\right) = \sigma\left(\sum_{k=1}^{K} \theta_k \phi_{c_k}(x) + \theta_0\right)$ where $\sigma$ is the sigmoid function and $\boldsymbol{\theta} = (\theta_0, \cdots, \theta_K)$ are the parameters. We refer to $\mathbf{c}$ as the "support" of the model. Throughout, we use capital and lowercase letters to differentiate between random variables and realized values, respectively. For instance, $C$ refers to a random concept whereas $c$ refers to a particular one. We generally used boldface to denote vectors and regular font to denote scalars, with the exception of the model input of each example being denoted with regular font.

### 3.1   Review: Bayesian variable selection

Learning CBMs shares many similarities with the more standard problem of learning sparse models in high-dimensional settings, where one must select among a finite number of features that have already been tabulated [49–51]. Here, we review the Bayesian solution for this more standard setting and then discuss how it may be extended to the CBM setting.

Given priors on $\mathbf{c}$ and $\boldsymbol{\theta}$, the goal of Bayesian inference for sparse models in high-dimensional settings is to sample from the posterior $p(\boldsymbol{\theta}, \mathbf{c}|\mathbf{y}, \mathbf{X})$, which allows for uncertainty quantification of the true support and coefficients. Posterior samples also describe the uncertainty at a new point $x_0$ via the posterior predictive distribution $p(y_0|x_0, \mathbf{y}, \mathbf{X}, \mathbf{c}) = \iint p(y_0|x_0, \boldsymbol{\theta}, \mathbf{c})p(\boldsymbol{\theta}, \mathbf{c}|\mathbf{y}, \mathbf{X})d\boldsymbol{\theta}d\mathbf{c}$, which can be viewed as combining the posterior samples into an ensemble model. Factorizing the posterior per $p(\boldsymbol{\theta}, \mathbf{c}|\mathbf{y}, \mathbf{X}) = p(\boldsymbol{\theta}|\mathbf{c}, \mathbf{y}, \mathbf{X})p(\mathbf{c}|\mathbf{y}, \mathbf{X})$, the first term describes the posterior for the model parameters and the second describes the posterior over selected variables. As the former can be readily obtained by classical Bayesian inference for low-dimensional settings (e.g., posterior inference for LR coefficients), our discussion will focus on the latter.

Inference for $p(\mathbf{c}|\mathbf{y}, \mathbf{X})$ is typically achieved through Gibbs sampling. For each outer loop, Gibbs rotates through indices $k = 1, \ldots, K$, during which it replaces the $k$th concept in the current iterate $\mathbf{c}$ by drawing from the posterior distribution for $C_k$ conditional on the other concepts $\mathbf{c}_{-k}$. When one is unable to sample from the conditional posterior, Metropolis-within-Gibbs can be used instead [52]. Rather than immediately replacing the $k$th concept, it proposes a candidate concept $\check{c}_k$ given the other concepts per some distribution $Q(C_k; \mathbf{C}_{-k} = \mathbf{c}_{-k})$ and accepts the candidate concept with probability $\alpha$. This acceptance probability $\alpha$, delineated in Algorithm 1, depends on the relative posterior probabilities of the candidate and existing concepts and is carefully designed so that the posterior distribution is stationary with respect to the sampling procedure.

We extend Bayesian variable selection to learn CBMs, where the key difference is that the number of potential concepts is effectively infinite, which makes it difficult to formulate an appropriate prior over concepts $p(\mathbf{C})$ and proposal transition kernel $Q(C_k; \mathbf{C}_{-k})$. Intuitively, it seems that LLMs can help perform both these tasks, but how to do so correctly and efficiently is surprisingly subtle. To motivate our choice, we first describe two naïve options:

---

**Algorithm 1** Metropolis-within-Gibbs

1: Initialize concept set $\mathbf{c} = (c_1, \cdots, c_K)$
2: List of concept sets $L = []$
3: **for** $t = 1, 2, \ldots, T$ **do**
4:     **for** $k = 1, 2, \ldots, K$ **do**
5:         $c_k \leftarrow$ MH-UPDATE($\mathbf{c}, k$)    // Update $k$-th concept
6:         Append the current $\mathbf{c}$ to $L$
7: **return** $L$
8: **function** MH-UPDATE($\mathbf{c}, k$)
9:     Propose concept $\check{c}_k \sim Q(C_k; \mathbf{c}_{-k})$
10:     $\alpha \leftarrow$ $\min\left\{ \frac{p((\mathbf{c}_{-k}, \check{c}_k)|\mathbf{y}, \mathbf{X})Q(c_k; \mathbf{c}_{-k})}{p(\mathbf{c}|\mathbf{y}, \mathbf{X})Q(\check{c}_k; \mathbf{c}_{-k})}, 1 \right\}$
11:     **if** Accept with probability $\alpha$ **then return** $\check{c}_k$
12:     **else return** $c_k$

---

*Update 1: Propose with the LLM's "prior."* Given the working concepts $\mathbf{C}_{-k} = \mathbf{c}_{-k}$, we can prompt the LLM to propose new concepts, without revealing any data, e.g. "Propose an additional concept for predicting readmissions, given the already-existing concepts $\mathbf{c}_{-k}$". [3] This can be viewed as treating the proposal distribution $Q(C_k; \mathbf{C}_{-k} = \mathbf{c}_{-k})$ as sampling from the LLM's conditional prior distribution $p(C_k|\mathbf{c}_{-k})$. Consequently, the MH acceptance ratio simplifies to $\frac{p(\mathbf{y}|(\mathbf{c}_{-k}, \check{c}_k), \mathbf{X})}{p(\mathbf{y}|\mathbf{c}, \mathbf{X})}$, which allows us to override proposed concepts that lead to poor model performance. Without access to data however, the LLM is likely to propose many irrelevant concepts from its prior, resulting in low acceptance probabilities and slow sampling. So while Update 1 may provide valid inference, it is inefficient and costly.

*Update 2: Propose with the LLM's "posterior."* Leveraging LLMs' ability to perform ICL [46, 47], another idea is to include all the data when prompting the LLM to propose new concepts. That is, we ask the LLM itself to "conduct posterior inference" and interpret the LLM's proposal distribution $Q(C_k; \mathbf{C}_{-k} = \mathbf{c}_{-k})$ as sampling from a conditional posterior distribution $p(C_k|\mathbf{c}_{-k}, \mathbf{y}, \mathbf{X})$ for some implicitly defined concept prior. This approach is risky; while LLMs may be able to approximate posterior inference for small datasets, this approximation breaks down for larger sample sizes [48]. Furthermore, we can no longer override poor concepts with data as the MH acceptance ratio simplifies to 1, i.e. the proposed concept is *always* accepted. So while Update 2 may generate concepts more efficiently, the sampling procedure may not converge to a good or even valid posterior distribution.

These update options are at the end of two extremes, trading off between efficient sampling versus unbiased posterior inference. Our task then is to find a better balance.

---

[3] If one believes the LLM's prior knowledge about the outcome of interest may be misleading, we can mask this information, e.g. "Propose an additional concept for predicting some label $Y$, given the already-existing concepts $\mathbf{c}_{-k}$."

## 3.2 The Split-sample update: Propose with the LLM's *partial* posterior

Rather than relying on the LLM's prior or posterior, a key idea in BC-LLM is to take the middle road by sampling from an LLM's *partial posterior*, which we will show enjoys the best of both worlds. The proposal, which we refer to as *Split-sample Metropolis update* (Algorithm 2), uses the LLM to propose candidate concepts based on *only a portion of the data* and fixes mistakes in the LLM's proposal using the remaining data in the accept-reject step. This addresses the drawbacks of the two naïve updating procedures in Section 3.1, as it simultaneously uses the data to generate more efficient proposals while overriding the LLM as needed.

More specifically, the split-sample update first randomly splits the data to construct subset $S$ of size $\lfloor \omega n \rfloor$ and its complement $S^c$, for some fraction $\omega$ (e.g. half). The LLM is prompted to generate candidate concepts by combining its prior knowledge with information from the data in $S$. We can then interpret the LLM's proposal distribution $Q(C_k; \mathbf{C}_{-k} = \mathbf{c}_{-k})$ as the (conditional) partial posterior distribution $p(C_k | \mathbf{c}_{-k}, \mathbf{y}_S, \mathbf{X})$. Then in the accept-reject step, the MH acceptance probability simplifies to the *partial* Bayes factor $\frac{p(\mathbf{y}_{S^c} | \mathbf{y}_S, (\mathbf{c}_{-k}, \check{c}), \mathbf{X})}{p(\mathbf{y}_{S^c} | \mathbf{y}_S, \mathbf{c}, \mathbf{X})}$ [53]. This compares the likelihood of the held-out data $S^c$ with respect to the candidate and existing concepts—much like sample-splitting in frequentist settings—and thus provides an opportunity to correct inconsistencies between the LLM's proposal and the actual posterior distribution.

---

**Algorithm 2** Split-sample Metropolis update

1: **function** SS-MH-UPDATE($\mathbf{c}$, $k$)
2:     Sample subset $S$ of size $\lfloor \omega n \rfloor$.
3:     Propose candidate $\check{c} \sim Q(C_k; \mathbf{c}_{-k}, \mathbf{y}_S, \mathbf{X})$
4:     $\alpha \leftarrow \min\left\{ \frac{p(\mathbf{y}_{S^c} | \mathbf{y}_S, (\mathbf{c}_{-k}, \check{c}), \mathbf{X})}{p(\mathbf{y}_{S^c} | \mathbf{y}_S, \mathbf{c}, \mathbf{X})}, 1 \right\}$
5:     **if** Accept with probability $\alpha$ **then return** $\check{c}$
6:     **else return** $c_k$

---

**Algorithm 3** Multiple-try split-sample Metropolis update

1: **function** MULTI-SS-MH-UPDATE($\mathbf{c}$, $k$)
2:     Sample subset $S$ of size $\lfloor \omega n \rfloor$.
3:     Propose $\check{c}_k^{(1)}, ..., \check{c}_k^{(M)} \sim Q(C_k; \mathbf{c}_{-k}, \mathbf{y}_S, \mathbf{X})$
4:     Sample $\check{m} \in \{1, ..., M\}$ with probabilities $\propto w_m \leftarrow p(\mathbf{y}_{S^c} | \mathbf{y}_S, (\mathbf{c}_{-k}, \check{c}^{(m)}), \mathbf{X}) Q(\check{c}^{(m)}; \mathbf{c}_{-k}, \mathbf{y}_S, \mathbf{X})$
5:     $w_0 \leftarrow p(\mathbf{y}_{S^c} | \mathbf{y}_S, \mathbf{c}, \mathbf{X}) Q(c_k; \mathbf{c}_{-k}, \mathbf{y}_S, \mathbf{X})$
6:     $\alpha \leftarrow \min\left\{ \frac{Q(c_k; \mathbf{c}_{-k}, \mathbf{y}_S, \mathbf{X}) \sum_{m=1}^{M} w_m}{Q(\check{c}_k^{(\check{m})}; \mathbf{c}_{-k}, \mathbf{y}_S, \mathbf{X}) \sum_{0 \leq m \leq M, m \neq \check{m}} w_m}, 1 \right\}$
7:     **if** Accept with probability $\alpha$ **then return** $\check{c}_k^{(\check{m})}$
8:     **else return** $c_k$

---

In practice, rather than sampling a single candidate concept, we use a version of multiple-try Metropolis-Hastings (Algorithm 3) [54]. Here, the LLM proposes a *batch* of candidates. We sample a candidate from this batch based on their posterior probabilities and consider the sampled candidate for acceptance. This is more cost-effective as it lets us batch concept annotations, as described later.

Assuming that the LLM's partial posterior across all iterations is consistent with some prior distribution $p(\mathbf{C})$, we can prove that the Markov chain defined by this procedure has the posterior $p(\mathbf{c} | \mathbf{y}, \mathbf{X})$ as its stationary distribution (see Proposition H.1 in the Appendix). Since LLMs are known to be imperfect Bayesian inference engines [48], this assumption is at best only approximately satisfied. Nevertheless, we can prove that the procedure still converges to the true concepts *even when this assumption does not hold*:

**Theorem 3.1.** *Suppose the data is IID. Let* $L(\mathbf{c}) \coloneqq \max_{\boldsymbol{\theta}} \mathbb{E}\{\log p(Y | X, \boldsymbol{\theta}, \mathbf{c})\}$ *and* $\mathcal{C}^* \coloneqq \arg\max_{\mathbf{c}} L(\mathbf{c})$. *For sample size* $n$, *let* $\Pi_n$ *denote the set of stationary distributions of the Markov chain defined by running Algorithm 1 with* SS-MH-UPDATE *or* MULTI-SS-MH-UPDATE *instead of* MH-UPDATE. *Assuming a Gaussian prior for* $\boldsymbol{\theta}$ *and under regularity conditions (see Assumption G.1), then* $\inf\{\pi(\mathcal{C}^*) \colon \pi \in \Pi_n\} \to 1$ *in probability as* $n \to \infty$.

## 3.3 Putting it together: BC-LLM

We now put all the pieces together. Mathematically, BC-LLM runs Metropolis-within-Gibbs (Algorithm 1) with a split-sample update (Algorithm 2 or 3). We translate this into a step-by-step recipe (Fig 1): After initializing BC-LLM in Step 0, each iteration of Metropolis-within-Gibbs (Steps 1 to 4) performs a split-sample update to output a CBM. The CBMs form a posterior distribution, as well as an ensemble prediction model. Example prompts and implementation details are in the Appendix.

**Step 0: Initialization and keyphrase extraction** We first use an LLM to extract "keyphrases" from each observation $x$, which will be used by the LLM to brainstorm candidate concepts. Here, "keyphrases" refer to short phrases: keyphrases for images describe what's in the image (e.g. blue eyes), while keyphrases for text data can be direct quotes or summarizations (e.g. diabetes) [55]. Keyphrases can be thought of as rough concepts, as they do not define a function but can help an LLM come up with a formal concept question (e.g. "diabetes" is a keyphrase whereas "Does the patient currently have diabetes?" is a concept). This brainstorming phase does not restrict the set of keyphrases the LLM can extract to allow for a wide exploration of concepts. The tradeoff is that the extracted keyphrases may be an incomplete summary of each observation. To initialize the CBM sampling procedure, these keyphrases are used to fit a (penalized) LR model for the target $Y$, similar to fitting a bag-of-words (BoW) model. An LLM then analyzes the most predictive keyphrases to generate an initial set of $K$ concepts and extracts these concepts from each observation.

**Step 1: Drop a concept & fit a keyphrase model** Each iteration proposes candidate concepts based on some data subset $S$ to replace the $k$th concept for $k \in \{1, \cdots, K\}$. Given the context limits of LLMs, having an LLM review each observation in $S$ individually is ineffective. Instead, we generate a summary by highlighting the top keyphrases in $S$ for predicting $Y$, on top of the existing $K - 1$ concepts $\mathbf{c}_{-k}$. For continuous outcomes, this can be accomplished by finding the top keyphrases associated with the residuals of a model that predicts $Y$ given $\mathbf{c}_{-k}$. For binary/categorical outcomes, we instead fit a (multinomial) LR model of $Y$, where the inputs are keyphrases encoded using BoW and annotated concepts $\mathbf{c}_{-k}$. Let the coefficients in this "keyphrase model" $f$ for keyphrases and concepts be denoted $\boldsymbol{\beta}_W$ and $\boldsymbol{\beta}_{\mathbf{c}_{-k}}$, respectively. We fit it by solving $\min_{\boldsymbol{\beta}_{\mathbf{c}_{-k}}, \boldsymbol{\beta}_W} \frac{1}{n} \sum_{i=1}^{n} \ell\left(y_i, f\left(x_i | \boldsymbol{\beta}_{\mathbf{c}_{-k}}, \boldsymbol{\beta}_W\right)\right) + \lambda J(\boldsymbol{\beta}_W)$, where $\ell$ is the logistic loss, $\boldsymbol{\beta}_W$ is penalized with some penalty function $J$, and penalty parameter $\lambda$ is selected by cross-validation. We found the ridge penalty to work well in practice but other penalties may be used instead.

**Step 2: Query the LLM for candidate concepts** Given the summary of the top keyphrases generated from Step 1, the LLM is asked to propose $M$ candidates for the $k$-th concept. In addition, we ask the LLM to provide the value of the partial posterior probability for each candidate concept. For instance, if the top keyphrases include "chronic kidney disease" and "acute kidney injury," the LLM may propose a concept like "Does the patient have a history of kidney disease?" with an assigned partial posterior probability of 0.4.

**Step 3: Concept annotation** Given candidate concepts, we use the LLM to extract their values using a zero-shot approach, i.e. $\phi_c(x)$ for $c \in \{\check{c}_k^{(1)}, \ldots, \check{c}_k^{(M)}\}$. Because *multiple-try* Metropolis-within-Gibbs proposes a batch of candidate concepts, we can batch their extractions into a single LLM query for each observation. This makes BC-LLM more cost/query-efficient than ordinary Metropolis-within-Gibbs, which outputs only a single concept per iteration and thus cannot be batched. Note that letting the LLM occasionally output probabilities when it is unsure can be helpful, though it is critical to avoid information leakage (e.g. no joint training of concept extraction and label prediction).

**Step 4: Accept/reject step** The final step is to compute the sampling probabilities for the candidate concepts and the acceptance ratio, which involve comparing the proposal probabilities for the existing and candidate concept sets $\mathbf{c}$ as well as their split-sample posteriors $p(\mathbf{y}_{S^c} | \mathbf{y}_S, \mathbf{c}, \mathbf{X}) = \int p(\mathbf{y}_{S^c} | \boldsymbol{\theta}, \mathbf{c}, \mathbf{X}) p(\boldsymbol{\theta} | \mathbf{y}_S, \mathbf{c}, \mathbf{X}) d\boldsymbol{\theta}$. To estimate the integral, one approach is to sample from the posterior on $\boldsymbol{\theta}$, but this can be computationally expensive since one would need to implement posterior sampling in inner and outer loops. Alternatively, in the case of LR-based CBMs, one can use a Laplace-like approximation [56] (see Appendix). As the posterior distribution of LR parameters are asymptotically normal, this approximation becomes increasingly accurate as $n \to \infty$.

**Computational cost.** BC-LLM performs $O(nTK)$ LLM queries, where $T$ is the number of outer loops. $K$ is typically chosen to be small, because a large $K$ yields less interpretable CBMs. Although standard Bayesian procedures typically draw thousands of posterior samples, we found that a small $T$ was quite effective in practice when combined with a greedy warm-start procedure; our experiments all use $T = 5$. Generally speaking, a small $T$ suffices for achieving high prediction accuracy, a larger $T$ (e.g., 10) is helpful for quantifying the uncertainty of relevant concepts. Critically, running BC-LLM is significantly more cost-effective than standard CBMs, which require $O(nW)$ queries where $W$ is the number of pre-specified concepts and must be large (typically hundreds or thousands) to avoid missing relevant concepts [8]. In our experiments, BC-LLM was quick to run: each iteration completes within one to two minutes for the experiments in Section 4.

Table 1: Performance of CBMs and black-box models (ResNet) for classifying bird species.

| Method | In-distribution | | | OOD |
| | Accuracy (↑) | AUC (↑) | Brier (↓) | Entropy (↑) |
| --- | --- | --- | --- | --- |
| BC-LLM | **0.680** (0.614, 0.747) | **0.874** (0.840, 0.907) | **0.428** (0.357, 0.500) | 0.865 (0.693, 1.036) |
| LLM+CBM | 0.640 (0.573, 0.707) | 0.810 (0.768, 0.853) | 0.452 (0.377, 0.528) | 0.663 (0.474, 0.852) |
| Boosting LLM+CBM | 0.538 (0.463, 0.614) | 0.722 (0.673, 0.772) | 0.577 (0.499, 0.654) | 0.842 (0.630, 1.054) |
| Human+CBM | 0.658 (0.591, 0.725) | 0.835 (0.791, 0.879) | 0.499 (0.414, 0.584) | 0.758 (0.558, 0.959) |
| LLM+CBM (No keyphrases) | 0.555 (0.488, 0.623) | 0.759 (0.713, 0.805) | 0.651 (0.548, 0.754) | 0.626 (0.495, 0.757) |
| ResNet | 0.664 (0.613, 0.716) | 0.853 (0.821, 0.885) | 0.457 (0.398, 0.516) | **0.914** (0.748, 1.079) |

## 4 Experiments

We evaluated BC-LLM in across three domains and modalities: classifying birds in images (Section 4.1), simulated outcomes from clinical notes (Section 4.2), and readmission risk in real-world clinical data (Section 4.3). The experiments below used GPT-4o-mini [57] and Section 4.3 used a protected health information-compliant version of GPT-4o for real-world clinical notes. The Appendix includes results using other LLMs, implementation details, and experimental settings.

**Baselines.** We compared against CBMs fit using the standard approach where humans select and annotate concepts (Human+CBM) [5], CBMs where an LLM brainstorms concepts after seeing which keyphrases are most associated with the label but without any iteration (LLM+CBM) [58, 10], and CBMs fit using a boosting algorithm that iteratively adds concepts by having an LLM analyze observations with large residuals (Boosting LLM+CBM) [17, 27]. We include black-box and semi-interpretable models to assess the tradeoff between interpretability and accuracy. For the bird-classification dataset, we also compare to a CBM that uses LLM-suggested concepts rather than keyword-based concepts; we use the 370 LLM-suggested concepts from [8] and extract the values of these concepts with an LLM (LLM+CBM (No keyphrases)).

**Evaluation metrics.** Methods were evaluated with respect to *predictive performance* as measured by AUC and/or accuracy and *uncertainty quantification* as measured by Brier score. When true concepts are known, we also quantify *concept selection rates* using "concept precision," as defined by $\frac{1}{K}\sum_{k=1}^{K} p(\hat{C}_k \in \mathbf{c}^* \mid \mathbf{y}, \mathbf{X})$, and "concept recall," as defined by $\frac{1}{K}\sum_{k=1}^{K} p(c_k^* \in \hat{\mathbf{C}} \mid \mathbf{y}, \mathbf{X})$, where $\hat{\mathbf{C}}$ is the posterior distribution over concept sets from BC-LLM or a single set of concepts learned by a non-Bayesian procedure. Concept matches were verified manually (see Appendix for details).

### 4.1 CBMs for classifying bird images

We first evaluate how well BC-LLM can learn concepts that differentiate between bird species within the same family in the standard CUB-birds image dataset [59]. Grouping the 200 bird species into their respective families, this resulted in 37 prediction tasks. We considered family-based prediction tasks because there likely exists a parsimonious CBM within each bird family, in contrast to the task of predicting between all 200 species. To assess how well BC-LLM performs in settings with limited prior knowledge, we do *not* tell the LLM that we are predicting bird species. That is, we replace the labels with one-hot encodings and simply tell the LLM to find concepts for predicting $Y$. Consequently, the LLM must search over a larger space of concepts.

The data is split 50/50 between training and testing. The number of concepts $K$ learned for each task was set to the number of classes, but no smaller than 4 and no greater than 10. For the black-box comparator, we fine-tuned the last layer of ResNet50 pre-trained on ImageNetV2 [60, 61]. Human+CBM was trained on the 312 human-annotated features available in the CUB-birds dataset.

BC-LLM achieved higher accuracy and better calibration on average, compared to all the other fully interpretable CBMs (Table 1). Notably, soft CBMs no longer performed as well once the concepts were constrained to be fully interpretable. BC-LLM also outperformed CBMs trained on human collated and annotated features, potentially because it was able to consider additional concepts. More critically, BC-LLM outperformed ResNet50 in both accuracy and calibration, as ResNet50 easily overtrained on the small bird datasets. In contrast, BC-LLM could identify relevant concepts for a simple parametric model, leading to faster model convergence.

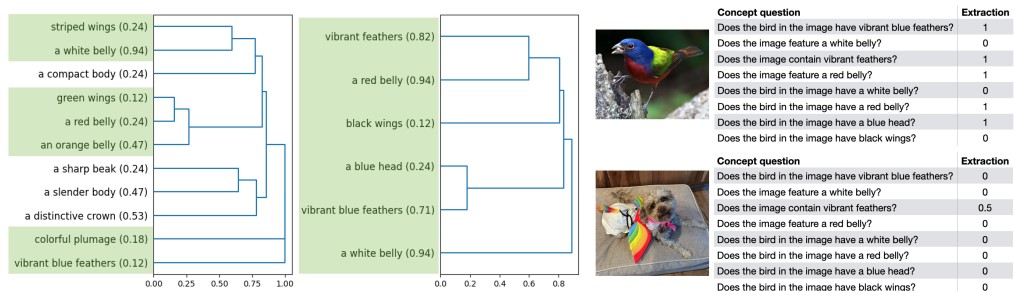

Figure 2: Example of BC-LLM classifying Bunting birds. (Left) Learned concepts trained on 1/3 versus 3/3 of the training data, respectively. Labels are shortened concept questions, generally of the form "Does the image depict...?" Proportion of posterior samples with the concept are shown in parentheses. Concepts are clustered hierarchically solely for visualization purposes. Highlighted labels are distinguishing bird features. (Right) Application of BC-LLM to an actual bunting bird (top) versus a dog pretending to be one (bottom).

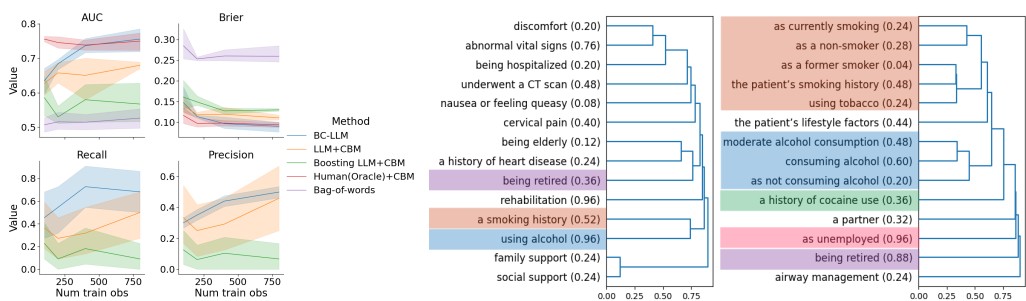

Figure 3: MIMIC results: (Left) Comparison of BC-LLM and existing methods in terms of performance and recovery of true concepts with 95% CI. (Right) Dendrograms of concepts learned by BC-LLM with 100 and 800 observations (left and right, respectively). Labels are shortened concept questions generally of the form "Does the note mention the patient...?" Highlighted labels correspond to the true color-coded concepts in Section 4.2.

We also assessed BC-LLM's robustness in OOD settings, as the Bayesian framing should improve uncertainty quantification (Table 1). For each model, we used it to classify birds for a species it was not trained on. Compared to other CBM methods, BC-LLM was more appropriately unsure, yielding higher entropy across the output classes. We note that ResNet model had the highest entropy, though it has the advantage of not being limited to extracting a small set of interpretable concepts. To illustrate the benefit of using fully interpretable CBMs in OOD settings, Figure 2 (right) shows the difference when applying the Bunting bird CBM to an actual bunting bird versus an OOD sample. Whereas the CBM extracted zeros and ones for the actual bunting bird, it was unsure how to extract the concept "Does this image show vibrant feathers?" for the image of a dog wearing rainbow wings. The LLM thus outputted a probability of 0.5, citing that "the image contains vibrant colors but the wings do not actually contain feathers." This shows how having an LLM output its reasoning when answering concrete concept questions can reveal situations that need human intervention.

Finally, we visualize the posterior distribution over concepts using hierarchical clustering (Fig 2 left). Continuing with the Bunting bird example, we see that BC-LLM converges towards distinguishing concepts as training data increases and away from vague or non-specific concepts (e.g. "sharp beaks" apply to all bunting birds). Moreover, the number of unique concepts decreases with the amount of training data, because it has converged to concepts that truly distinguish the different species.

## 4.2 CBMs for classifying clinical notes with simulated outcomes

To objectively measure how well BC-LLM can recover true concepts, we conducted a simulation study where the true concepts are known. To assess BC-LLM in settings where the LLM has no prior

knowledge about the target, we simulated $Y$ using a LR model with five Social Determinants of Health as inputs, which were annotated from real-world patient notes in MIMIC-IV [62, 63]:

**1.** Does the note mention the patient consuming alcohol in the present or the past?
**2.** Does the note mention the patient smoking in the present or the past?
**3.** Does the note mention the patient using recreational drugs in the present or the past?
**4.** Does the note imply the patient is unemployed?
**5.** Does the note imply the patient is retired?

CBMs with $K$=6 were trained on 100 to 800 observations. `Human(Oracle)+CBM` uses true concepts.

BC-LLM outperforms the comparator methods both in predictive performance and calibration (Figure 3a) and, with only 400 observations, performs as well as `Human(Oracle)+CBM`. The performance gap between BC-LLM and the comparator methods widens with more training observations because BC-LLM can iteratively refine its concepts. Although `Boosting LLM+CBM` is also an iterative procedure, it struggles to hypothesize relevant concepts because it reviews only a few misclassified observations each iteration and cannot revise concepts added in previous iterations. Additionally, BC-LLM was better at recovering the true concepts compared to existing methods.

Visualizing the posterior distribution using hierarchical clustering (Fig 3 right), the learned posterior notably contracts towards the true concepts with increasing data as the number of training observations increases from 100 to 800. Moreover, BC-LLM remains appropriately unsure about certain concepts that are highly correlated. For instance, it states many variations of the smoking concept that are essentially impossible to distinguish between, which is both expected and desired.

### 4.3 Augmenting a tabular model with clinical notes

Here we demonstrate how BC-LLM can help ML developers integrate different data modalities to improve model performance. The data science team at Zuckerberg San Francisco General Hospital has both tabular data and unstructured notes from the electronic health record (EHR). The team has so far fit a model using the tabular data to predict readmission risk for heart failure patients and wants to assess if clinical notes contain additional concepts that are predictive. To address this, we extend BC-LLM to take as inputs (i) the risk prediction from the original tabular model and (ii) $K = 4$ concepts extracted from the clinical notes, which can be viewed as revising the existing model by adding features. The CBMs were trained on 1000 patients and evaluated on 500 held-out patients.

The original tabular model, trained on lab and flowsheet values, achieved an AUC of 0.60. BC-LLM revised this model to perform substantially better, achieving an AUC of 0.64 (Fig 4 left). Moreover, it outperforms comparator methods with respect to both AUC and brier score.

To assess the interpretability of the learned concepts, five clinicians were asked to rate concepts in the CBMs as well as top features in the original tabular model by how predictive they were, from 1=low to 3=high clinical relevance.[4] Concepts from BC-LLM received an average rating of 2.5, whereas concepts from the other methods were rated $\leq 2$ (Fig 4 right). When unblinded to the concepts, clinicians commented on a number of advantages to using BC-LLM. First, many of the learned concepts were known to be causally relevant but were difficult to engineer solely from tabular data. For example, information for determining "substance dependence" is rarely documented in the tabular data but often noted in clinical notes. Second, BC-LLM also suggested new features to engineer that the data science team had not previously considered. Most interestingly, some of the learned concepts pointed to interventions that the hospital could try to implement, such as ensuring follow-up appointments were scheduled and opting for medication regimes that were easier to follow. In contrast, the top features in the original tabular model were difficult to interpret and not actionable.

## 5 Discussion

BC-LLM is a new approach to learning CBMs that iteratively proposes concepts using an LLM within a Bayesian framework, which allows for rigorous uncertainty quantification despite LLMs

---

[4]Tabular features were rewritten as questions so they could not be distinguished from learned concepts. See Appendix for annotation instructions.

| Method | AUC (95% CI) | Brier (95% CI) |
|---|---|---|
| BC-LLM | **0.64** (0.58, 0.70) | **0.14** (0.12, 0.62) |
| LLM+CBM | 0.59 (0.52, 0.65) | 0.29 (0.25, 0.33) |
| Boosting LLM+CBM | 0.59 (0.52, 0.66) | **0.14** (0.12, 0.16) |
| Bag-of-words | 0.52 (0.46, 0.58) | 0.29 (0.25, 0.34) |

Figure 4: Learning to augment a readmission risk prediction model for heart failure patients. Dendrogram labels are shortened questions of the format "Does the note mention the patient having...?". Highlighted concepts received scores from clinicians as being highly predictive (scores 2.5+). Average clinician ratings for concepts/features from the different methods are shown on the right.

being prone to error and hallucinations. Because it explores and suggests concepts in a data-adaptive manner, BC-LLM is particularly well-suited for settings with limited prior knowledge about which concepts are relevant or where the number of potentially relevant concepts is infinite. The method is compatible with various data modalities (text, images, and tabular data) and can be extended beyond the settings of binary and multiclass classification. The empirical results show that BC-LLM outperforms existing methods, even black-box models in certain settings.

*Future work.* BC-LLM is currently designed for learning highly interpretable CBMs where the number of concepts $K$ is typically no more than $20$. While BC-LLM can be applied to learn even more concepts, future directions can consider further speeding up posterior inference, such as through mini-batching of observations or concepts.

*Impact Statement.* The interpretability provided by BC-LLM additionally enables better understanding of the underlying algorithm and facilitates human oversight, which may improve the safety of AI algorithms and AI-based decision-making. The use of LLMs incurs significant computational cost and corresponding environmental impacts; however, BC-LLM is much more computationally efficient than comparable LLM-based CBMs, potentially reducing overall environmental impacts.

## Acknowledgments

This work was funded through a Patient-Centered Outcomes Research Institute® (PCORI®) Award (ME-2022C1-25619). The views presented in this work are solely the responsibility of the author(s) and do not necessarily represent the views of the PCORI®, its Board of Governors or Methodology Committee. The authors thank the UCSF AI Tiger Team, Academic Research Services, Research Information Technology, and the Chancellor's Task Force for Generative AI for their technical support related to the use of Versa API gateway. The authors thank Patrick Vossler, Yifan Mai, Seth Goldman, Andrew Bishara, James Marks, Julian Hong, Aaron Kornblith, Nicholas Petrick, and Gene Pennello for their invaluable feedback on the work.

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

# Appendix

## A  Multiple-Try Metropolis-Hastings

In this section, we provide intuition for the Multiple-Try Metropolis-Hastings partial posterior method (Algorithm 3). To draw connections with the original Multiple-Try Metropolis-Hastings method [54], we first work in a completely abstract setting.

Given a state space $\Omega$, stationary distribution $\pi$, and proposal transition kernel $T(y; x)$, define weights $w(x, y) = \pi(x)T(y; x)$. Note that under regular Metropolis-Hastings, the acceptance ratio for proposing the state $y$ given a state $x$ is $\min\{w(y, x)/w(x, y), 1\}$.

**Classical version.**  We briefly describe [54]'s original method. Suppose the current state is $x$.

1. Draw $z_1, z_2, \ldots, z_M \sim T(Z; x)$.
2. Sample $y$ from a distribution with probabilities

$$\mathbb{P}\{Y = z_j\} = \frac{w(z_j, x)}{\sum_{i=1}^{M} w(z_i, x)}. \tag{1}$$

   Let $\check{m}$ denote the sampled index.
3. Sample $z_1^*, \ldots, z_{M-1}^* \sim T(Z; y)$, and set $z_M^* = x$.
4. Accept $y$ with probability

$$\beta(x, y; \mathbf{z}, \mathbf{z}^*) = \min\left\{\frac{\sum_{i=1}^{M} w(z_i, x)}{\sum_{i=1}^{M} w(z_i^*, y)}, 1\right\}. \tag{2}$$

One can show that this Markov chain satisfies the detailed balance equations when marginalizing out the intermediate states $z_1, z_2, \ldots, z_M, z_1^*, \ldots, z_M$ (see Theorem 1 in [54]). We would like, however, to minimize the number of proposals necessary, so it is natural to ask whether the algorithm can be modified such that the proposals needed for the backward transitions, $z_1^*, \ldots, z_{M-1}^*$, can be omitted. This is indeed possible.

**Modified version.**

1. Draw $z_1, z_2, \ldots, z_M \sim T(Z; x)$.
2. Sample $y$ from a distribution with probabilities

$$\mathbb{P}\{Y = z_j\} = \frac{w(z_j, x)}{\sum_{i=1}^{M} w(z_i, x)}. \tag{3}$$

   Let $\check{m}$ denote the sampled index, $\mathbf{z} = (z_1, z_2, \ldots, z_M)$, $\mathbf{z}^* = (z_1, \ldots, z_{\check{m}-1}, x, z_{\check{m}+1}, \ldots, z_M)$, and set $q(y; \mathbf{z}) = \mathbb{P}\{Y = y | z_1, \ldots, z_M\}$.
3. Accept $y$ with probability

$$\begin{aligned}\alpha(x, y; \mathbf{z}_{-\check{m}}) &= \min\left\{\frac{p(y \to \mathbf{z}^* \to x)}{p(x \to \mathbf{z} \to y)}, 1\right\} \\ &= \min\left\{\frac{\pi(y)\prod_{i=1}^{M} T(z_i; y)q(x; \mathbf{z}^*)}{\pi(x)\prod_{i=1}^{M} T(z_i^*; x)q(y; \mathbf{z})}, 1\right\}.\end{aligned} \tag{4}$$

**Proposition A.1.** *The modified Multiple-Try MH sampler is reversible.*

*Proof.* Let $\Phi(y; x)$ denote the actual transition matrix. We want to show

$$\pi(x)\Phi(y; x) = \pi(y)\Phi(x; y). \tag{5}$$

To see this, we compute

$$\pi(x)\Phi(x, y) = k\pi(x)T(y; x)\int \prod_{i=2}^{M} T(z_i; x)q(y; \mathbf{z})\alpha(x, y; \mathbf{z}_{-1})d\mathbf{z}_{-1}$$

$$= k\int \min\left\{\pi(y)T(x; y)\prod_{i=2}^{M} T(z_i; y)q(x; \mathbf{z}^*), \pi(x)T(y; x)\prod_{i=2}^{M} T(z_i; x)q(y; \mathbf{z})\right\}d\mathbf{z}_{-1}$$

$$= \pi(y)\Phi(y, x).$$

$\square$

**Connections between modified and classical versions.** The acceptance probability in the modified version of Multiple-Try MH can be expanded as follows:

$$\min\left\{\frac{\pi(y)\prod_{i=1}^{M} T(z_i; y)q(x; \mathbf{z}^*)}{\pi(x)\prod_{i=1}^{M} T(z_i^*; x)q(y; \mathbf{z})}, 1\right\} = \min\left\{\frac{\pi(y)\prod_{i=1}^{M} T(z_i; y)\frac{w(x,y)}{\sum_{i=1}^{M} w(z_i^*, y)}}{\pi(x)\prod_{i=1}^{M} T(z_i^*; x)\frac{w(y,x)}{\sum_{i=1}^{M} w(z_i, x)}}, 1\right\}$$

$$= \min\left\{\frac{\sum_{i\neq\tilde{m}} w(z_i, x) + w(y, x)}{\sum_{i\neq\tilde{m}} w(z_i, y) + w(x, y)} \cdot \prod_{i=2}^{M}\frac{T(z_i; y)}{T(z_i; x)}, 1\right\}. \tag{6}$$

If $T(-, -)$ is invariant in the first argument (as in our application to BC-LLM), then this last formula is exactly equal to $\beta(x, y; \mathbf{z}, \mathbf{z}^*)$, the acceptance probability for the classical version, but if we were to use the same points for both the non-realized forward and backward proposals.

**Equivalence with MULTI-SS-MH-UPDATE.** Finally, we show that the modified version of Multiple-Try MH described in the previous sections is equivalent to what is implemented in MULTI-SS-MH-UPDATE, for a fixed $S$. To see this, we make the replacements:

- $x \leftarrow \mathbf{c}$
- $z_i \leftarrow (\check{c}_k^{(i)}, \mathbf{c}_{-k})$ for $i = 1, \ldots, M$
- $y \leftarrow (\check{c}_k^{(\tilde{m})}, \mathbf{c}_{-k})$
- $T(-; -) \leftarrow Q(-; -, \mathbf{y}_S, \mathbf{X})$
- $\pi(-) \leftarrow p(-|\mathbf{y}, \mathbf{X})$

Plugging these into (6), the acceptance ratio becomes

$$\min\left\{\frac{Q(c_k; \mathbf{c}_{-k}, \mathbf{y}_S, \mathbf{X})\sum_{m=1}^{M} p((\check{c}^{(m)}, \mathbf{c}_{-k})|\mathbf{y}, \mathbf{X})}{Q(\check{c}_k^{(\tilde{m})}; \mathbf{c}_{-k}, \mathbf{y}_S, \mathbf{X})\sum_{m=1}^{M} p((\check{c}^{(m)}, \mathbf{c}_{-k})|\mathbf{y}, \mathbf{X})}, 1\right\}$$

$$= \min\left\{\frac{Q(c_k; \mathbf{c}_{-k}, \mathbf{y}_S, \mathbf{X})\sum_{m=1}^{M} p(\mathbf{y}_{S^c}|\mathbf{y}_S, (\check{c}_k^{(m)}, \mathbf{c}_{-k}), \mathbf{X})Q(\check{c}_k^{(m)}; \mathbf{c}_{-k}, \mathbf{y}_S, \mathbf{X})}{Q(\check{c}_k^{(\tilde{m})}; \mathbf{c}_{-k}, \mathbf{y}_S, \mathbf{X})\sum_{0\leq m\leq M, m\neq\tilde{m}} p(\mathbf{y}_{S^c}|\mathbf{y}_S, (\check{c}_k^{(m)}, \mathbf{c}_{-k}), \mathbf{X})Q(\check{c}_k^{(m)}; \mathbf{c}_{-k}, \mathbf{y}_S, \mathbf{X})}, 1\right\}, \tag{7}$$

where the equality comes from (44) and we set $\check{c}_k^{(0)} = c_k$ for convenience of notation. Observe that this is exactly the formula in Line 6 of Algorithm 3.

Meanwhile, plugging into (3) and using (44), we see that the sampling weights used in the modified version of Multiple-Try MH are equivalent to those in Line 3 of Algorithm 3. This completes the proof of the equivalence.

# B Implementation details for BC-LLM

Here we discuss how the hyperparameters for BC-LLM should be selected:

- **Fraction $\omega$ of data used for partial posterior**: Choosing a small $\omega$ may lead to the LLM proposing less relevant concepts, but tends to lead to more diverse proposals. In contrast, a large $\omega$ tends to lead to less diverse proposals because the LLM is encouraged to propose concepts that are relevant to the dataset $\mathcal{D}$, which may not necessarily generalize. In experiments, we found that $\omega = 0.5$ provided good results.
- **Number of candidate concepts $M$**: More candidate proposals per iteration can allow for more efficient exploration of concepts. The number of candidates is limited by the number of concepts that the LLM can reliably extract in a single batch. There tends to also be diminishing returns, as the first few candidates generated by the LLM based on the top keyphrases tend to be highest quality and most relevant. We found that setting $M = 10$ provided good performance.
- **Warm-start and Burn-in**: Since Gibbs sampling can be slow to converge, we precede it with a warm-start, which we obtain by updating concepts greedily. That is, we select the concept that maximizes $\mathrm{argmax}\, p(\gamma | \mathbf{c}_{-k}, \mathbf{y}_S, \mathbf{X})$, instead of sampling from the distribution. In experiments, we run warm-start for one epoch and stored the last 20 iterates as posterior samples; the rest of the samples were treated as burn-in.
- **Number of iterations $T$**: Although Monte Carlo procedures for Bayesian inference typically have thousands of samples, this is cost-prohibitive when an LLM must query each observation for a new concept per iteration. In experiments, we found that even setting $T$ as low as 4 to be quite effective. Generally speaking, if the goal is solely prediction accuracy, a small $T$ may suffice. On the other hand, if the goal is uncertainty quantification for the relevant concepts, one may prefer $T$ on the higher end (e.g. 10) for more complete exploration of concepts.

## C  Example prompts

Here we provide example prompts that one may use with BC-LLM. The prompt should vary with how much prior knowledge one has about the prediction target as well as how much information one would like to reveal to the LLM about the prediction target.

### C.1  Example prompt for keyphrase extraction (Step 0)

Example prompt for extracting keyphrases from images, where we reveal limited information to the LLM about the prediction target:

```
 Given an image, your task is to brainstorm a set of
descriptors that will help classify the image to its
corresponding label Y. For the provided image, list as
many descriptors about it.  For each descriptor, also
list as many descriptors that mean the same thing or
generalizations of the descriptor.  All descriptors, synonyms,
and generalizations cannot be more than two words.  Output
characteristics that this image possess, such as "round
wings" or "red head." Do not output general categories of
descriptors like "wing color" or "head shape." Output at
least 10 keyphrases.
```

Example prompt for extracting keyphrases from clinical notes, where we do not reveal information about the prediction target but we provide ideas on patient characteristics that one should extract:

```
 Here is a clinical note:  ⟨ note ⟩
Output a list of descriptors that summarizes the patient
case (such as aspects on demographics, diagnoses, social
determinants of health, etc).  For each descriptor, also
list as many descriptors that mean the same thing or
generalizations of the descriptor.  All descriptors, synonyms,
and generalizations cannot be more than two words.  Output as
a JSON in the following format...
```

## C.2 Example prompt for proposing concepts (Step 2)

Example prompt for having the LLM brainstorm candidate concepts based on top keyphrases in the keyphrase model, where we do not reveal information about the prediction target and only reveal that the dataset contains patient notes:

```
 The goal is to come up with a concept bottleneck model (CBM)
that only extracts 3 meta-concepts from patient notes to
predict some outcome Y with maximum accuracy.  A meta-concept
is a binary feature extractor defined by a yes/no question.
We have 2 meta-concepts so far:
1. ...
2. ...
To come up with the 3rd meta-concept, I have done the
following:  I first fit a CBM on the 2 existing meta-concepts.
Then to figure out how to improve this 3-concept CBM, I first
asked an LLM to extract a list of concepts that are present
in each note, and then fit a linear regression model on the
extracted concepts to predict the residuals of the 3-concept
CBM. These are the top extracted concepts in the resulting
residual model, in descending order of importance:
⟨ top keyphrases from keyphrase model ⟩
Given the residual model, create cohesive candidates for the
3rd meta-concept.  Be systematic and consider all the listed
concepts in the residual model.  Start from the most to the
least predictive concept.  For each concept, check if it
matches an existing meta-concept or create a new candidate
meta-concept.  Work down the list, iterating through each
concept.  Clearly state each candidate meta-concept as a
yes/no question.
Suggestions for generating candidate meta-concepts:  Do
not propose meta-concepts that are simply a union of two
different concepts (e.g.  ''Does the note mention this
patient experiencing stomach pain or being female?" is not
allowed), questions with answers that are almost always a
yes (e.g.  the answer to ''Does the note mention this patient
being sick?" is almost always yes), or questions where the
yes/no options are not clearly defined (e.g.  ''Does the note
mention this patient experiencing difficulty?" is not clearly
defined because difficulty may mean financial difficulty,
physical difficulties, etc).  Do not propose meta-concepts
where you would expect over 95% agreement or disagreement
with the 4 existing meta-concepts (e.g.  ''Does the note
mention the patient having high blood pressure?" overlaps
too much with ''Does the note mention the patient having
hypertension?").
Finally, summarize all the generated candidate concepts in a
JSON.
```

## C.3 Example prompt for annotations for candidate concepts (Step 3)

A simple concept extraction procedure is to obtain binary or categorical concept extractions. We use prompts like this:

```
You will be given a clinical note.  I will give you a series
of questions.  Your task is answer each question with 1 for
yes or 0 for no.  If the answer to the question is not clearly
yes or no, you may answer with the probability that the answer
is a yes.  Respond with a JSON that includes your answer to
all of the questions.  Questions:
```

```
1. Does the note mention the patient having social support?
2. Does the note mention the patient having stable blood
   pressure?
3. Does the note mention the patient not experiencing
   respiratory distress?
4. Does the note mention the patient experiencing substance
   abuse or dependence?
5. Does the note mention the patient being uninsured?

clinical note: ⟨ note ⟩
```

Note that in cases where the LLM was unsure about the concept's value, it was allowed to return a probability, which prior works have found to be helpful [6]. In general, the LLM outputted 1's and 0's. Probabilities were outputted occasionally, which as highlighted in the main manuscript, can be audited and checked by a human.

## D    Additional Results

### D.1    MIMIC

To assess the robustness of BC-LLM, we include the following variations. First, we test the robustness of BC-LLM to prompt phrasing. In particular, we feed in a prompt that is more vague than that used in the main manuscript during Step 0 of BC-LLM by replacing the original sentence "Output a list of descriptors that summarizes the patient case (such as aspects on demographics, diagnoses, SDOH, etc)." with "Output a list of descriptors that summarizes the patient case." Second, we test the robustness of the results to the choice of LLM by rerunning the same experiment using Cohere's Command-R LLM, rather than GPT-4o-mini.

Results are shown in Figure 5. First, we see that BC-LLM is quite robust to prompt phrasing, as the results with this simpler prompt are similar to the more detailed prompt. Second, we see that performance of all the methods—BC-LLM and the comparator methods—are lower using Command-R. Nevertheless, we see that the performance rankings between the methods are similar, with BC-LLM performing better for nearly all metrics.

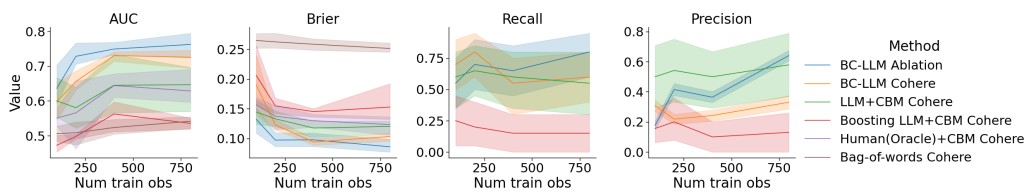

Figure 5:   Additional results running BC-LLM and comparator methods on the MIMIC dataset, evaluated in terms of performance and recovery of true concepts. Error bars indicate the 95% CI.

### D.2    CUB-Birds dataset

Table 2 reports results from two additional sets of experiments on the CUB-Birds dataset. First, to evaluate robustness of BC-LLM to the choice of LLM, we report the performance of BC-LLM and comparator methods when Anthropic's Claude 3.5 Haiku is used instead, with all other settings the same as that discussed in Section 4.1. Second, to facilitate comparisons to other "hard concept" CBM results on the CUB-Birds dataset, we evaluate BC-LLM on the dataset's original task of classifying all 200 bird species, setting $K = 100$. Because the size of this dataset and the number of concepts is substantially larger, we only ran the greedy phase of BC-LLM. Nevertheless, BC-LLM outperforms the comparator methods.

| Method | Accuracy (↑) | AUC (↑) | Brier (↓) |
|---|---|---|---|
| *Using Claude 3.5 Haiku* | | | |
| BC-LLM | 0.665 (0.593, 0.736) | 0.849 (0.811, 0.888) | 0.443 (0.367, 0.519) |
| LLM+CBM | 0.583 (0.517, 0.648) | 0.771 (0.722, 0.821) | 0.507 (0.434, 0.580) |
| Boosting LLM+CBM | 0.640 (0.568, 0.712) | 0.811 (0.769, 0.852) | 0.552 (0.455, 0.649) |
| *Classifying between all 200 bird species* | | | |
| BC-LLM | 0.558 (0.514, 0.600) | 0.992 (0.990, 0.994) | 0.593 (0.562, 0.625) |
| LLM+CBM | 0.516 (0.474, 0.558) | 0.988 (0.984, 0.991) | 0.606 (0.567, 0.646) |
| Boosting LLM+CBM | 0.005 (0.003, 0.007) | 0.500 (0.500, 0.500) | 1.012 (1.011, 1.012) |

Table 2: Additional results for the CUB-Birds dataset

## D.3 Other image datasets

To evaluate BC-LLM on other real-world image datasets, Table 3 includes two additional imaging datasets:

- *Functional Map of the World (fMoW)* [64]: RGB satellite images classified into 62 building/land use categories. Models were trained on satellite images from the US, with 100 images sampled from each category. CBMs were trained to have 60 concepts. Models were evaluated in terms of in-distribution performance (i.e. images from the USA) as well as out-of-distribution (OOD) performance when evaluated on satellite images from China.
- *Imagenette* [65]: 10 classes from ImageNet. CBMs were trained to have 10 concepts.

| Method | Accuracy (↑) | AUC (↑) | Brier (↓) |
|---|---|---|---|
| *fMoW: USA images* | | | |
| BC-LLM | 0.357 (0.340, 0.375) | 0.904 (0.898, 0.910) | 0.780 (0.766, 0.792) |
| LLM+CBM | 0.311 (0.295, 0.327) | 0.878 (0.871, 0.885) | 0.892 (0.872, 0.914) |
| Boosting LLM+CBM | 0.118 (0.105, 0.130) | 0.757 (0.748, 0.764) | 1.029 (1.018, 1.039) |
| *fMoW: OOD-China images* | | | |
| BC-LLM | 0.265 (0.246, 0.285) | 0.840 (0.830, 0.849) | 0.853 (0.839, 0.868) |
| LLM+CBM | 0.223 (0.205, 0.242) | 0.791 (0.780, 0.801) | 1.023 (1.00, 1.045) |
| Boosting LLM+CBM | 0.084 (0.072, 0.098) | 0.707 (0.693, 0.722) | 1.065 (1.054, 1.077) |
| *Imagenette* | | | |
| BC-LLM | 0.987 (0.980, 0.993) | 0.999 (0.997, 1.000) | 0.072 (0.062, 0.085) |
| LLM+CBM | 0.902 (0.883, 0.919) | 0.988 (0.986, 0.990) | 0.125 (0.107, 0.142) |
| Boosting LLM+CBM | 0.639 (0.610, 0.670) | 0.899 (0.888, 0.909) | 0.448 (0.422, 0.479) |

Table 3: Results on additional imaging datasets

# E Experiment details

## E.1 Evaluating correct concept matches

Quantifying a "correct" concept match requires going beyond exact string matching, as there are many possible refinements for a single concept. For instance, for the concept "Does the note mention the patient smoke at present?", there are rephrasings ("Does this note mention if the patient currently smokes?"), polar opposites ("Does the note not mention the currently smoking?"), or closely overlapping concepts ("Does the note mention the patient's smoking history?"). To quantify if a learned concept was correct, we used the following rule: if the absolute value of the Pearson correlation for LLM annotations of two concepts exceeds 50%, the two concepts are sufficiently similar. We manually confirmed that any pair of learned and true concepts that passed this rule were always closely related.

## E.2 MIMIC

The entire dataset consists of 7043 observations of which we trained on 100, 200, 400, and 800 randomly selected observations and evaluated 500 held-out observations. We used the "Chief Complaint" and "Social History" sections of the notes. The label $Y$ was generated per a LR model, in which

$$\log \frac{\Pr(Y = 1|X)}{\Pr(Y = 0|X)} = 4 * \mathbb{1}\{\text{Does the note imply the patient is unemployed?}\}$$
$$+ 4 * \mathbb{1}\{\text{Does the note imply the patient is retired?}\}$$
$$+ 4 * \mathbb{1}\{\text{Does the note mention the patient consuming alcohol in the present or the past?}\}$$
$$- 4 * \mathbb{1}\{\text{Does the note mention the patient smoking in the present or the past?}\}$$
$$+ 5 * \mathbb{1}\{\text{Does the note mention the patient using recre- ational drugs in the present or the past?}\}$$

`LLM+CBM summarization` and BC-LLM were run to fit CBMs with $K = 6$ concepts. BC-LLM was run for $T = 5$ epochs. `Boosting LLM+CBM` ran for 10 iterations, in which at most 1 new candidate was added each iteration.

## E.3 CUB-Birds dataset

To train a bird classifier for $R$ subtypes, `LLM+CBM summarization` and BC-LLM were run to fit CBMs with $K = \min(10, \max(4, R))$ concepts. BC-LLM was run for a maximum of $T = 5$ epochs or 25 iterations, whichever was reached first. `Boosting LLM+CBM` ran for 10 iterations, in which at most 1 new candidate was added each iteration. For the black-box comparator, we used a ResNet50 model pre-trained on ImageNetV2, which ensures ResNet's training set does not overlap with the CUB-Birds dataset.

## E.4 Clinical notes and tabular data from the Zuckerberg San Francisco General Hospital

Data in this experiment contained PHI, for which IRB approval was obtained. To predict readmission risk, the models were trained to analyze the sections "Brief history leading to hospitalization" and the "Summary of hospitalization" in the discharge summary for each patient. `LLM+CBM summarization` and BC-LLM were run to fit CBMs with $K = 4$ concepts. BC-LLM was run for $T = 5$ epochs. `Boosting LLM+CBM` ran for 10 iterations, in which at most 1 new candidate was added each iteration.

## E.5 Clinician survey

To conduct the survey for clinical relevance of features and concepts, the following instructions were given to clinicians:

```
Please rate the features learned by BC-LLM and its comparator methods.  In
the attached CSV, we have listed 25 or so features from various methods,
where we mask which method has learned which feature.  Please fill in the
column "How clinically relevant is the candidate concept with determining
a patient's readmission risk?  Enter values = 1:  low, 2:medium, 3:  high."
Please try to use the full range of scores, rather than assigning all the
features the same score.
```

Table 4 shows the table that clinicians were asked to fill in. The ordering of features was randomly shuffled to obfuscate which method had learned which concepts.

# F Laplace Approximation of Split-Sample Posterior

In this section, we describe how to perform Laplace approximation of the split-sample posterior $p(\mathbf{y}_{S^c}|\mathbf{y}_S, \mathbf{c}, \mathbf{X})$ when the likelihood model is logistic and the prior on the coefficient vector $\boldsymbol{\theta}$ is a standard normal $\mathcal{N}(0, \gamma^2 I)$. For simplicity, we omit discussing the constant term $\theta_0$. Fixing the list of concepts $\mathbf{c}$, we let $\boldsymbol{\Phi}$ denote the $n \times K$ matrix whose $(i, j)$-th entry is given by $\Phi_{ij} = \phi_{c_j}(x_i)$.

Table 4: Clinicians were asked to fill in the following table with their ratings of the clinical relevance of each feature/concept for predicting a patient's readmission risk.

| Candidate concept | Score (1-3) |
|---|---|
| Does the patient have multiple medications prescribed? | |
| Does the patient have a history of hypertension (HTN)? | |
| Is this patient experiencing chest pain? | |
| What is the patient's weight? | |
| Does the patient have heart failure with reduced ejection fraction? | |
| What are the patient's lab results for BNP? | |
| Does the patient have substance dependence? | |
| Is the patient stable at discharge? | |
| Does the patient have a history of falls? | |
| How many emergency department encounters does this patient have? | |
| What are the patient's lab results for Lactate Dehydrogenase? | |
| Does the patient have uncontrolled diabetes? | |
| Does the patient have a history of frequent hospital admissions? | |
| How many encounters does this patient have? | |
| Does the patient have a history of diabetes mellitus type 2 (DM2)? | |
| Does the patient have complex medical history? | |
| Does the patient have a history of sepsis? | |
| What are the patient's lab results for Creatinine? | |
| Does the patient have a history of drug or substance use disorder? | |
| What is the patient's value for Expiratory Positive Airway Pressure (EPAP)? | |
| Does the patient have had any missed outpatient appointments? | |
| What are the patient's lab results for Glucose? | |
| Are the patient's outpatient follow-up appointments scheduled? | |

Given any subset of example indices $T \subset [n]$, we can write the joint likelihood for these examples as

$$p(\mathbf{y}_T | \boldsymbol{\theta}, \mathbf{c}, \mathbf{X}) = \exp\left(\sum_{i \in T} \left(y_i \boldsymbol{\theta}^T \boldsymbol{\Phi}_{i\cdot} - \log(1 + \exp(\boldsymbol{\theta}^T \boldsymbol{\Phi}_{i\cdot}))\right)\right). \tag{8}$$

Multiplying by the prior for $\boldsymbol{\theta}$, the joint conditional distribution for $\mathbf{y}_T$ and $\boldsymbol{\theta}$ is then:

$$p(\boldsymbol{\theta}, \mathbf{y}_T | \mathbf{c}, \mathbf{X}) = (2\pi\gamma^2)^{-1/2} \exp\left(\sum_{i \in T} \left(y_i \boldsymbol{\theta}^T \boldsymbol{\Phi}_{i\cdot} - \log(1 + \exp(\boldsymbol{\theta}^T \boldsymbol{\Phi}_{i\cdot}))\right) - \frac{\|\boldsymbol{\theta}\|_2^2}{2\gamma^2}\right). \tag{9}$$

To form the Laplace approximation to (9), we first denote

$$g_T(\boldsymbol{\theta}) := -\log p(\boldsymbol{\theta}, \mathbf{y}_{S^c} | \mathbf{c}, \mathbf{X})$$
$$= \sum_{i \in T} \left(-y_i \boldsymbol{\theta}^T \boldsymbol{\Phi}_{i\cdot} + \log(1 + \exp(\boldsymbol{\theta}^T \boldsymbol{\Phi}_{i\cdot}))\right) + \frac{\|\boldsymbol{\theta}\|_2^2}{2\gamma^2}, \tag{10}$$

and its Hessian by

$$\mathbf{H}_T(\boldsymbol{\theta}) := \nabla^2 g_T(\boldsymbol{\theta}) = \sum_{i \in T} \frac{e^{\boldsymbol{\theta}^T \boldsymbol{\Phi}_{i\cdot}}}{(1 + e^{\boldsymbol{\theta}^T \boldsymbol{\Phi}_{i\cdot}})^2} \boldsymbol{\Phi}_{i\cdot} \boldsymbol{\Phi}_{i\cdot}^T + \gamma^{-2} \mathbf{I}. \tag{11}$$

Let

$$\boldsymbol{\theta}_{\mathrm{MAP},T} := \arg\min g_T(\boldsymbol{\theta}) \tag{12}$$

be the maximum a posteriori (MAP) estimate. We can then perform a quadratic approximation of the exponent around $\boldsymbol{\theta}_{\mathrm{MAP},T}$ to get

$$p(\boldsymbol{\theta}, \mathbf{y}_T | \mathbf{c}, \mathbf{X}) \approx p(\boldsymbol{\theta}_{\mathrm{MAP},T}, \mathbf{y}_T | \mathbf{c}, \mathbf{X}) \exp\left(-\frac{1}{2}(\boldsymbol{\theta} - \boldsymbol{\theta}_{\mathrm{MAP},T})^T \mathbf{H}_T(\boldsymbol{\theta}_{\mathrm{MAP},T})(\boldsymbol{\theta} - \boldsymbol{\theta}_{\mathrm{MAP},T})\right). \tag{13}$$

Plugging in (9) evaluated at $\boldsymbol{\theta}_{\mathrm{MAP},T}$ and integrating out with respect to $\boldsymbol{\theta}$ gives

$$
\begin{aligned}
p(\mathbf{y}_T|\mathbf{c},\mathbf{X}) \approx \exp &\left(\sum_{i\in T}\left(y_i\boldsymbol{\theta}_{\mathrm{MAP},T}^T\boldsymbol{\Phi}_{i\cdot}-\log(1+\exp(\boldsymbol{\theta}_{\mathrm{MAP},T}^T\boldsymbol{\Phi}_{i\cdot}))\right)+\frac{\|\boldsymbol{\theta}_{\mathrm{MAP},T}\|_2^2}{2\gamma^2}\right)\\
&\cdot(2\pi)^{(K-1)/2}\gamma^{-1}\det(\mathbf{H}_T(\boldsymbol{\theta}_{\mathrm{MAP},T}))^{-1/2}.
\end{aligned}
\tag{14}
$$

We finish by applying the above formulas with respect to $T=[n]$ and $T=S$ and taking ratios:

$$
\begin{aligned}
p(\mathbf{y}_{S^c}|\mathbf{y}_S,\mathbf{c},\mathbf{X}) &= \frac{p(\mathbf{y}|\mathbf{c},\mathbf{X})}{p(\mathbf{y}_S|\mathbf{c},\mathbf{X})}\\
&\approx\exp\left(\sum_{i=1}^n\left(y_i\boldsymbol{\theta}_{\mathrm{MAP}}^T\boldsymbol{\Phi}_{i\cdot}-\log(1+\exp(\boldsymbol{\theta}_{\mathrm{MAP}}^T\boldsymbol{\Phi}_{i\cdot}))\right)\right.\\
&\qquad\left.-\sum_{i\in S}\left(y_i\boldsymbol{\theta}_{\mathrm{MAP},S}^T\boldsymbol{\Phi}_{i\cdot}-\log(1+\exp(\boldsymbol{\theta}_{\mathrm{MAP},S}^T\boldsymbol{\Phi}_{i\cdot}))\right)\right)\\
&\qquad\cdot\exp\left(\frac{\|\boldsymbol{\theta}_{\mathrm{MAP},S}\|_2^2-\|\boldsymbol{\theta}_{\mathrm{MAP}}\|_2^2}{2\gamma^2}\right)\left(\frac{\det(\mathbf{H}_S(\boldsymbol{\theta}_{\mathrm{MAP},S}))}{\det(\mathbf{H}(\boldsymbol{\theta}_{\mathrm{MAP}}))}\right)^{1/2}.
\end{aligned}
\tag{15}
$$

*Remark* F.1. The MAP estimate $\boldsymbol{\theta}_{\mathrm{MAP}}$ can be computed by a call to `scikit-learn`'s `LogisticRegression()` model, setting $\mathtt{C}=2\gamma^2$.

*Remark* F.2. In the current implementation of our procedure, we use a slightly different method to perform Laplace approximation, which instead approximates the integral $p(\mathbf{y}_{S^c}|\mathbf{y}_S,\mathbf{c},\mathbf{X})=\int p(\mathbf{y}_{S^c}|\boldsymbol{\theta},\mathbf{c},\mathbf{X})p(\boldsymbol{\theta}|\mathbf{y}_S,\mathbf{c},\mathbf{X})d\boldsymbol{\theta}$.

# G  Proof of Theorem 3.1

Recall the definition

$$
L(\mathbf{c}):=\max_{\boldsymbol{\theta}}\mathbb{E}_{(X,Y)\sim\nu}\{\log p(Y|X,\boldsymbol{\theta},\mathbf{c})\}.
\tag{16}
$$

This quantifies how relevant the concepts in $\mathbf{c}$ are to the prediction of the response $Y$. Note that the definition does not require that the model class is correctly specified (i.e. there does not have to be some $\mathbf{c}$ and $\boldsymbol{\theta}$ so that $p(Y=1|X=x,\boldsymbol{\theta},\mathbf{c})=\sigma\left(\sum_{k=1}^K\theta_k\phi_{c_k}(x)+\theta_0\right)$). Before presenting the proof, we first detail the assumptions we make.

**Assumption G.1.**

  i. $Y\in\{0,1\}$ is a binary response;
  ii. The data $\mathcal{D}$ is generated i.i.d. from some distribution $\nu$;
  iii. The prior on $\boldsymbol{\theta}$ is $\mathcal{N}(0,\gamma^2\mathbf{I})$;
  iv. $\mathcal{C}^*:=\mathrm{argmax}_{\mathbf{c}}L(\mathbf{c})$ comprises all permutations of a single vector $\mathbf{c}^*$;
  v. There exists some $\Delta>0$ such that $L(\mathbf{c}^*)-\mathrm{argmax}_{\mathbf{c}\notin\mathcal{C}^*}L(\mathbf{c})\geq\Delta$;
  vi. Let $\mathcal{C}_n:=\cup_{\mathbf{c}_{-k},\mathbf{y},\mathbf{X}}\{c:Q(c;\mathbf{c}_{-k},\mathbf{y},\mathbf{X})>0\}$. Assume $\mathcal{C}_n$ is a finite set of size at most $\exp(n^{1-\epsilon})$ for some $0<\epsilon<1$;
  vii. There exists $\eta>0$ such that for each $\mathbf{c}_{-k}$, there exists $l\in\{1,2,\ldots,K\}$ such that $c_l^*=\mathrm{argmax}_cL((c,\mathbf{c}_{-k}))$, $c_l^*\notin\mathbf{c}_{-k}$, and $Q(c_l^*;\mathbf{c}_{-k},\mathbf{y},\mathbf{X})\geq\eta$ almost surely for all $\mathbf{y},\mathbf{X}$ with $n$ large enough;
  viii. There exists some $B$ such that for any $\mathbf{c}\in\mathcal{C}_n^K$, all distinct, we have $\|\boldsymbol{\theta}_{\mathbf{c}}^*\|_2\leq R$, where

$$
\boldsymbol{\theta}_{\mathbf{c}}^*:=\mathrm{argmax}_{\boldsymbol{\theta}}\mathbb{E}_{(X,Y)\sim\nu}\{\log p(Y|X,\boldsymbol{\theta},\mathbf{c})\};
\tag{17}
$$

  ix. Let $\Phi_{\mathbf{c}}=\Phi_{\mathbf{c}}(X)=(\phi_{c_1}(X),\phi_{c_2}(X),\ldots,\phi_{c_K}(X))$ for $X\sim\nu$. There exists some $\lambda_{\min}$ such that for any $\mathbf{c}\in\mathcal{C}_{n,K}$, all eigenvalues of the second moment matrix $\mathbb{E}[\Phi_{\mathbf{c}}\Phi_{\mathbf{c}}^T]$ are bounded from below by $\lambda_{\min}$. Here, $\mathcal{C}_{n,K}$ denotes the $K$-fold product of $\mathcal{C}_n$ but with any vectors containing duplicate concepts removed;
  x. We modify the algorithm so that the random subset $S$ can only take one of $\exp(n^{1-\epsilon})$ possibilities $\mathcal{S}_n:=\{S_1,S_2,\ldots,S_{B,n}\}$.

Note that for simplicity, we avoid discussing the constant term $\theta_0$. Most components in Assumption G.1 are either standard regularity conditions ((i), (ii), (iv), (v), (vi), (viii), (ix)) that are often made

in statistical analysis or are hyperparameter choices ((iii) and (x)). The key new assumption is (vii), which asserts that at each step, there is at least a fixed nonzero probability of the LLM proposing one of the K "best" concepts conditional on the data. This helps to ensure at every outer loop of the sampler (i.e. for every cycle through all the positions of the concept vector), enough mass flows to the optimal set of concepts. We believe this is a reasonable assumption even in regimes where the LLM has limited knowledge, as long as the degree of knowledge is non-zero.

*Proof of Theorem 3.1.* We prove the theorem only for SS-MH-UPDATE for simplicity, but it will be clear how to generalize it to MULTI-SS-MH-UPDATE.

*Step 1: Taking a subsequence.* Note that the Markov chain in question corresponds to the list $L$ returned in line 7 of Algorithm 1. Denote this using $(\tilde{\mathbf{c}}^{\ t})_{t=1}^{\infty}$. Because of the cyclic nature of the updates, this chain is not time invariant and is hence difficult to work with. We argue that it suffices to study the temporal projection $(\mathbf{c}^{\ t})_{t=1}^{\infty}$, where $\mathbf{c}_t = \tilde{\mathbf{c}}_{tK}$ for $t = 0, 1, 2, \ldots$. This is because every stationary distribution for $(\tilde{\mathbf{c}}^{\ t})_{t=1}^{\infty}$ is also stationary for $(\mathbf{c}^{\ t})_{t=1}^{\infty}$.

*Step 2: Reduction to connected components.* Since $(\tilde{\mathbf{c}}^{\ t})_{t=1}^{\infty}$ satisfies the detailed balance equations, it is a symmetric Markov chain, i.e. it has an equivalent representation as a flow on a network, in which the vertices are states and the edges correspond to nonzero transition probabilities. This property is inherited by $(\mathbf{c}^{\ t})_{t=1}^{\infty}$. Consider a connected component $\mathcal{A}$ of the network. The Markov chain, when restricted to $\mathcal{A}$, is aperiodic, since there must exist a self-loop. Furthermore, it is finite according to Assumption G.1(vi). As such, $(\mathbf{c}^{\ t})_{t=1}^{\infty}$ has a unique stationary distribution $\pi_{\mathcal{A}}$ supported on $\mathcal{A}$ [66]. Indeed, all stationary distributions of $(\mathbf{c}^{\ t})_{t=1}^{\infty}$ are convex combinations of such distributions. Hence, it suffices to show the desired property for $\pi_{\mathcal{A}}$.

*Step 3: Concentration.* In order to analyze $\pi_{\mathcal{A}}$, we need quantitative control over the transition probabilities via concentration. For a given dataset $\mathcal{D}$ of size $n$, we condition on the event guaranteed by Proposition G.3.

*Step 4: Convergence.* We again focus on a single connected component $\mathcal{A}$ and the goal is to show that with high probability, $\pi_{\mathcal{A}}(\mathcal{C}^* \cap \mathcal{A}) \geq 1 - \delta$ for some $\delta \to 0$ uniformly. We may ignore the rest of the network. Let $a = |\mathcal{A}|$ and $b = |\mathcal{C}^* \cap \mathcal{A}|$. Since $\mathcal{A}$ itself is finite, we can enumerate its states starting with those in $\mathcal{C}^* \cap \mathcal{A}$. The stationary distribution can be written as a vector $\mathbf{v} = (\mathbf{v}_{1:b}, \mathbf{v}_{b+1:a})$, while the Markov chain itself can be represented as a transition matrix $P$, which we write in block form as

$$\mathbf{P} = \begin{bmatrix} \mathbf{P}_{11} & \mathbf{P}_{12} \\ \mathbf{P}_{21} & \mathbf{P}_{22} \end{bmatrix}. \tag{18}$$

The stationarity equation gives us

$$\mathbf{v}_{b+1:a}(\mathbf{I} - \mathbf{P}_{22}) = \mathbf{v}_{1:b}\mathbf{P}_{12}, \tag{19}$$

from which we obtain

$$\|\mathbf{v}_{b+1:a}\|_2 = \left\|\mathbf{v}_{1:b}\mathbf{P}_{12}(\mathbf{I} - \mathbf{P}_{22})^{-1}\right\|_2 \leq \frac{\|\mathbf{P}_{12}\|_{op}}{1 - \|\mathbf{P}_{22}\|_{op}}, \tag{20}$$

so long as the denominator on the right hand side is nonzero. To see this, notice that any state $j$ for $b + 1 \leq j \leq a$ corresponds to a vector $\tilde{\mathbf{c}}^{\ (0)} \notin \mathcal{C}^*$. However, using Assumption G.1(vii), there is a sequence $(\tilde{\mathbf{c}}^{\ (0)}, \tilde{\mathbf{c}}^{\ (1)}, \ldots, \tilde{\mathbf{c}}^{\ (K-1)})$ such that for $k = 1, \ldots, K$, we have (i) $\tilde{\mathbf{c}}_{-k}^{(k)} = \tilde{\mathbf{c}}_{-k}^{(k-1)}$, (ii) $Q(\tilde{c}_k; \tilde{\mathbf{c}}_{-k}^{(k-1)}, \mathbf{y}_S, \mathbf{X}) \geq \eta$, (iii) $L(\tilde{\mathbf{c}}^{\ (k)}) \geq L(\tilde{\mathbf{c}}^{\ (k-1)})$ with equality holding iff $\tilde{\mathbf{c}}^{\ (k)} = \tilde{\mathbf{c}}^{\ (k-1)}$. Combining (iii) with (28), we see that all acceptance ratios along this sequence are equal to 1, which means that the transition probability from state $j$ to states in $\mathcal{C}^*$ is at least

$$\prod_{k=1}^{K} Q(\tilde{c}_k; \tilde{\mathbf{c}}_{-k}^{(k-1)}, \mathbf{y}_S, \mathbf{X}) \geq \eta^K. \tag{21}$$

The sum of entries in row $j - b$ in $\mathbf{P}_{22}$ is the total probability of transition from state $j$ to other states within $\mathcal{A}\backslash\mathcal{C}^*$ and hence has value less than $1 - \eta^K$. This holds for any row. Applying Lemma G.2, we thus get $\|\mathbf{P}_{22}\|_{op} \leq 1 - \eta^K$, which allows us to bound the denominator in (20). Next, every entry in $\mathbf{P}_{12}$ is the probability of transition from a state in $\mathcal{C}^*$ to a state in $\mathcal{A}\backslash\mathcal{C}^*$. Using Step 3 and Assumption G.1(v), the acceptance probability $\alpha$ of such a transition satisfies

$$\log \alpha \leq \max_{\mathbf{c} \in \mathcal{A}\backslash\mathcal{C}^*} \log p(\mathbf{y}_{S^c}|\mathbf{y}_S, \mathbf{c}, \mathbf{X}) - \log p(\mathbf{y}_{S^c}|\mathbf{y}_S, \mathbf{c}^*, \mathbf{X})$$

$$\leq -n\Delta + O(n^{1-\epsilon'}). \tag{22}$$

We thus have

$$\|\mathbf{P}_{12}\|_{op} \leq \sqrt{(a-b)b} \exp\left(-n\Delta(1 + O(n^{-\epsilon'}))\right),\tag{23}$$

which allows us to bound the numerator in (20). Combining this with the earlier bound on the denominator gives

$$\|\mathbf{v}_{a+1:c}\|_2 \leq \sqrt{(a-b)b}\eta^{-K} \exp\left(-n\Delta(1 + O(n^{-\epsilon'}))\right).\tag{24}$$

We then compute

$$
\begin{aligned}
\pi_{\mathcal{A}}(\mathcal{C}^* \cap \mathcal{A}) &= \|\mathbf{v}_{1:b}\|_1 \\
&= 1 - \|\mathbf{v}_{b+1:a}\|_1 \\
&\geq 1 - \sqrt{a-b}\|\mathbf{v}_{a+1:c}\|_2 \\
&\geq 1 - \sqrt{b(a-b)^2}\eta^{-K} \exp\left(-n\Delta(1 + O(n^{-\epsilon'}))\right),
\end{aligned}\tag{25}
$$

whose error term we can bound as

$$
\begin{aligned}
\sqrt{b(a-b)^2}\eta^{-K} \exp\left(-n\Delta(1 + O(n^{-\epsilon'}))\right) &\lesssim |\mathcal{C}_n| \exp\left(-n\Delta(1 + O(n^{-\epsilon'}))\right) \\
&\leq \exp(n^{1-\epsilon}) \exp\left(-n\Delta(1 + O(n^{-\epsilon'}))\right) \\
&\leq \exp(O(n^{-\epsilon})).
\end{aligned}\tag{26}
$$

Finally, we note that all hidden constants can be chosen to be independent of $n$. □

**Lemma G.2.** *Let* $\mathbf{M}$ *be a* $m \times n$ *matrix whose entries are non-negative, and suppose* $\sum_{j=1}^n M_{ij} \leq \beta$ *for* $i = 1, 2, \ldots, m$. *Then* $\|\mathbf{M}\|_{op} \leq \beta$.

*Proof.* Let $\mathbf{D}$ be the $m \times m$ diagonal matrix whose entries are given by $D_{ii} = \sum_{j=1}^n M_{ij}$. Then $\mathbf{D}^{-1}\mathbf{M}$ is a stochastic matrix, which has operator norm at most 1, while $\|\mathbf{D}\|_{op} \leq \max_{1 \leq i \leq n} \sum_{j=1}^n M_{ij} \leq \beta$. Putting these together, we get

$$\|\mathbf{M}\|_{op} = \|\mathbf{D}\mathbf{D}^{-1}\mathbf{M}\|_{op} \leq \|\mathbf{D}\|_{op}\|\mathbf{D}^{-1}\mathbf{M}\|_{op} \leq \beta,\tag{27}$$

as we wanted. □

**Proposition G.3** (Concentration of split-sample posterior)**.** *Assuming the conditions in Assumption G.1, then for any* $\epsilon' < \epsilon/2$, *we have*

$$\sup_{\mathbf{c} \in \mathcal{C}_{n,K}, S \in \mathcal{S}_n} \left| \frac{1}{\lceil (1-\omega)n \rceil} \log p(\mathbf{y}_{S^c}|\mathbf{y}_S, \mathbf{c}, \mathbf{X}) - L(\mathbf{c}) \right| = O(n^{-\epsilon'})\tag{28}$$

*with probability converging to 1.*

*Proof.* Let us fix $\mathbf{c}$ for now. We first write

$$
\begin{aligned}
l_i(\boldsymbol{\theta}) &:= \log p(Y = y_i | X = \mathbf{x}_i, \boldsymbol{\theta}, \mathbf{c}) \\
&= y_i \boldsymbol{\theta}^T \boldsymbol{\Phi}_{i\cdot} - \log\left(1 + \exp(\boldsymbol{\theta}^T \boldsymbol{\Phi}_{i\cdot})\right).
\end{aligned}\tag{29}
$$

Let $\bar{l}(\boldsymbol{\theta}) := \mathbb{E}_{\mathbf{x}_i, y_i}[l_i(\boldsymbol{\theta})]$ and set $\boldsymbol{\theta}^* = \arg\max \bar{l}(\boldsymbol{\theta})$. For convenience, denote $\boldsymbol{\Phi} = \boldsymbol{\Phi}_{\mathbf{c}}$. Note that $\boldsymbol{\Phi}$ has entries bounded between 0 and 1 and hence has a second moment matrix with maximum eigenvalue bounded above by $K$. Furthermore, it is a sub-Gaussian random vector with bounded sub-Gaussian norm. Using these properties, we may find a universal value $R'$ such that $\mathbb{E}\left[\boldsymbol{\Phi}\boldsymbol{\Phi}^T \mathbf{1}(\|\boldsymbol{\Phi}\|_2 \leq R')\right] \succeq \frac{1}{2}\mathbb{E}\left[\boldsymbol{\Phi}\boldsymbol{\Phi}^T\right]$. Consider the ball $B(2R)$ of radius $2R$. For any $\boldsymbol{\theta} \in B(2R)$,

the Hessian of the population log likelihood hence satisfies

$$
\begin{aligned}
\mathbf{H}(\boldsymbol{\theta}) &:= \nabla^2 \bar{l}(\boldsymbol{\theta}) \\
&= -\mathbb{E}\left[\frac{\exp(\Phi^T \boldsymbol{\theta})}{(1 + \exp(\Phi^T \boldsymbol{\theta}))^2} \Phi \Phi^T\right] \\
&\preceq -\mathbb{E}\left[\frac{\exp(\Phi^T \boldsymbol{\theta})}{(1 + \exp(\Phi^T \boldsymbol{\theta}))^2} \Phi \Phi^T \mathbf{1}(\|\Phi\|_2 \le R')\right] \\
&\preceq -\frac{1}{2} \inf_{\boldsymbol{\theta} \in B(2R), \mathbf{v} \in B(R')} \frac{\exp(\mathbf{v}^T \boldsymbol{\theta})}{(1 + \exp(\mathbf{v}^T \boldsymbol{\theta}))^2} \mathbb{E}\left[\Phi \Phi^T\right] \\
&= \frac{e^{2R'R}}{2(1 + e^{2R'R})^2} \mathbb{E}\left[\Phi \Phi^T\right] \\
&\preceq -\frac{\lambda_{\min} e^{2R'R}}{2(1 + e^{2R'R})^2} I.
\end{aligned}
\tag{30}
$$

Using this lower bound on the curvature, we see that

$$
\bar{l}(\boldsymbol{\theta}^*) - \sup_{\boldsymbol{\theta} \in \partial B(2R)} \bar{l}(\boldsymbol{\theta}) \ge \frac{\lambda_{\min} R^2 e^{2R'R}}{2(1 + e^{2R'R})^2}.
\tag{31}
$$

Next, we use the fact that $n^{-1/2} \sum_{i=1}^n \left(l_i(\boldsymbol{\theta}) - \bar{l}(\boldsymbol{\theta})\right)$ is a sub-Gaussian process with increments bounded according to

$$
\left\| n^{-1/2} \sum_{i=1}^n \left(l_i(\boldsymbol{\theta}) - \bar{l}(\boldsymbol{\theta})\right) - n^{-1/2} \sum_{i=1}^n \left(l_i(\boldsymbol{\theta}') - \bar{l}(\boldsymbol{\theta}')\right) \right\|_{\psi_2} \lesssim K \|\boldsymbol{\theta} - \boldsymbol{\theta}'\|_2.
\tag{32}
$$

We can thus use Talagrand's comparison inequality [67] to get, for any $t > 0$, that

$$
n^{-1/2} \sup_{\boldsymbol{\theta} \in B(2R)} \left| \sum_{i=1}^n \left(l_i(\boldsymbol{\theta}) - \bar{l}(\boldsymbol{\theta})\right) \right| \lesssim R\sqrt{K} + Rt
\tag{33}
$$

with probability at least $1 - e^{-t^2}$. Take $t = n^{1/2 - \epsilon'}$ for any $\epsilon' < \epsilon/2$. Note that $\hat{l}_n(\boldsymbol{\theta}) := n^{-1} \sum_{i=1}^n l_i(\boldsymbol{\theta})$ is concave, with

$$
\hat{l}_n(\boldsymbol{\theta}^*) \ge \bar{l}(\boldsymbol{\theta}^*) - O(n^{-\epsilon'}),
\tag{34}
$$

while

$$
\begin{aligned}
\sup_{\boldsymbol{\theta} \in \partial B(2R)} \hat{l}_n(\boldsymbol{\theta}) &\le \sup_{\boldsymbol{\theta} \in \partial B(2R)} \bar{l}(\boldsymbol{\theta}) + O(n^{-\epsilon'}) \\
&\le \bar{l}(\boldsymbol{\theta}^*) + \frac{\lambda_{\min} R^2 e^{2R'R}}{2(1 + e^{2R'R})^2} + O(n^{-\epsilon'}).
\end{aligned}
\tag{35}
$$

As such, whenever $n$ is large enough, we see that $\hat{l}_n$ also attains its global maximum in $B(2R)$ and that this maximum value satisfies

$$
\left| \sup_{\boldsymbol{\theta} \in \mathbb{R}^K} \hat{l}_n(\boldsymbol{\theta}) - \bar{l}(\boldsymbol{\theta}^*) \right| = O(n^{-\epsilon'}).
\tag{36}
$$

Recall further that $\bar{l}(\boldsymbol{\theta}^*) = L(\mathbf{c})$. We next want to integrate out $p(\mathbf{y}|\boldsymbol{\theta}, \mathbf{c}, \mathbf{X}) = \exp(n\hat{l}_n(\boldsymbol{\theta}))$ with respect to the prior. Using (36), we get the upper bound

$$
\begin{aligned}
p(\mathbf{y}|\mathbf{c}, \mathbf{X}) &= \int p(\mathbf{y}|\boldsymbol{\theta}, \mathbf{c}, \mathbf{X}) \frac{1}{(2\pi\gamma^2)^{K/2}} \exp(-\|\boldsymbol{\theta}\|^2 / 2\gamma^2) d\boldsymbol{\theta} \\
&\le \exp(nL(\mathbf{c}) + O(n^{1-\epsilon'})).
\end{aligned}
\tag{37}
$$

To get a lower bound, we use the fact that the Hessian of the population likelihood also has an lower bound

$$\mathbf{H}(\boldsymbol{\theta}) \succeq -\|\mathbf{H}(\boldsymbol{\theta})\|_{op} I \succeq -\sqrt{K}\mathbf{I}. \tag{38}$$

For any $0 < r < R$, this gives a lower bound of

$$\inf_{\boldsymbol{\theta} \in B(\boldsymbol{\theta}^*, r)} L(\boldsymbol{\theta}) \geq L(\boldsymbol{\theta}^*) - \sqrt{K}r^2, \tag{39}$$

where $B(\boldsymbol{\theta}^*, r)$ is the ball of radius $r$ centered at $\boldsymbol{\theta}^*$. Using (33) again, we get

$$\inf_{\boldsymbol{\theta} \in B(\boldsymbol{\theta}^*, r)} \hat{l}_n(\boldsymbol{\theta}) \geq L(\boldsymbol{\theta}^*) - \sqrt{K}r^2 - O(n^{-\epsilon'}). \tag{40}$$

The prior density is bounded from below by $\frac{1}{(2\pi\gamma^2)^{K/2}} \exp(-2R^2/\gamma^2)$ on $B(2R)$. We therefore have

$$
\begin{aligned}
&\int p(\mathbf{y}|\boldsymbol{\theta}, \mathbf{c}, \mathbf{X}) \frac{1}{(2\pi\gamma^2)^{K/2}} \exp(-\|\boldsymbol{\theta}\|^2/2\gamma^2) d\boldsymbol{\theta} \\
&\geq \frac{r^K \operatorname{Vol}(B(1))}{(2\pi\gamma^2)^{K/2}} \exp(-2R^2/\gamma^2) \exp\left(n(L(\boldsymbol{\theta}^*) - \sqrt{K}r^2 - O(n^{-\epsilon'}))\right).
\end{aligned}
\tag{41}
$$

The maximum is achieved when we set $r = \frac{K^{1/4}}{(2n)^{1/2}}$. Combining (37) and (41) and taking logarithms, we get

$$\log p(\mathbf{y}|\mathbf{c}, \mathbf{X}) = nL(\mathbf{c}) + O(n^{1-\epsilon'}). \tag{42}$$

Performing the same calculation with $\mathbf{y}_S$ instead of $\mathbf{y}$ and taking differences, we get

$$\left| \frac{1}{\lceil (1-\omega)n \rceil} \log p(\mathbf{y}_{S^c}|\mathbf{y}_S, \mathbf{c}, \mathbf{X}) - L(\mathbf{c}) \right| = O(n^{-\epsilon'}). \tag{43}$$

Finally, we take a union bound over all $\mathbf{c} \in \mathcal{C}_{n,K}$ and $S \in \mathcal{S}_n$, noting that all hidden constants can be chosen to be independent of $\mathbf{c}$. $\qquad\square$

## H   Stationary Distribution Under Consistent Proposals

In this section, we motivate the multiple-try partial posterior Metropolis update (Algorithm 3) in more detail. Notably, we have the following proposition.

**Proposition H.1.** *Suppose the LLM proposes from the conditional partial posterior distribution, i.e. $Q(C_k; \mathbf{C}_{-k} = \mathbf{c}_{-k}) = p(C_k|\mathbf{c}_{-k}, \mathbf{y}_S, \mathbf{X})$. Then the Markov chain defined by running Gibbs sampling (Algorithm 1) with* SS-MH-UPDATE *or* MULTI-SS-MH-UPDATE *instead of* MH-UPDATE *has the posterior $p(\mathbf{C}|\mathbf{y}, \mathbf{X})$ as a stationary distribution.*

*Proof.* We first show this for SS-MH-UPDATE. It suffices to show that the acceptance probability in Line 4 of Algorithm 2 is equal to that of a Metropolis-Hastings filter with $p(\mathbf{C}|\mathbf{y}, \mathbf{X})$ as the target. We compute:

$$
\begin{aligned}
p((c, \mathbf{c}_{-k})|\mathbf{y}, \mathbf{X}) &= \frac{p(\mathbf{y}_{S^c}|\mathbf{y}_S, (c, \mathbf{c}_{-k}), \mathbf{X}) p(\mathbf{y}_S, (c, \mathbf{c}_{-k})|\mathbf{X})}{p(\mathbf{y}|\mathbf{X})} \\
&= p(\mathbf{y}_{S^c}|\mathbf{y}_S, (c, \mathbf{c}_{-k}), \mathbf{X}) p(c|\mathbf{c}_{-k}, \mathbf{X}, \mathbf{y}_S) \frac{p(\mathbf{c}_{-k}, \mathbf{y}_S|\mathbf{X})}{p(\mathbf{y}|\mathbf{X})}
\end{aligned}
\tag{44}
$$

Hence,

$$
\begin{aligned}
\frac{p((\check{c}, \mathbf{c}_{-k})|\mathbf{y}, \mathbf{X}) Q(c; \mathbf{c}_{-k}, \mathbf{y}_S, \mathbf{X})}{p((c, \mathbf{c}_{-k})|\mathbf{y}, \mathbf{X}) Q(\check{c}; \mathbf{c}_{-k}, \mathbf{y}_S, \mathbf{X})} &= \frac{p((\check{c}, \mathbf{c}_{-k})|\mathbf{X}, \mathbf{y}) p((c, \mathbf{c}_{-k})|\mathbf{y}, \mathbf{X})}{p((c, \mathbf{c}_{-k})|\mathbf{y}, \mathbf{X}) p((\check{c}, \mathbf{c}_{-k})|\mathbf{y}, \mathbf{X})} \\
&= \frac{p(\mathbf{y}_{S^c}|\mathbf{y}_S, (\mathbf{c}_{-k}, \check{c}), \mathbf{X})}{p(\mathbf{y}_{S^c}|\mathbf{y}_S, \mathbf{c}, \mathbf{X})}
\end{aligned}
\tag{45}
$$

as we wanted.

To show the statement for MULTI-SS-MH-UPDATE, we use its equivalence, for a fixed $S$, to the modified Multiple-Try Metropolis-Hastings method described in Appendix A. In Appendix A, we prove that this method satisfies the detailed balance equations for the posterior. Since the detailed balance equations are linear in the transition probabilities, we can marginalize this over $S$ to show that the posterior is stationary with respect for MULTI-SS-MH-UPDATE, which draws $S$ uniformly at random. $\qquad\square$

