# OpenReview forum: "Bayesian Concept Bottleneck Models with LLM Priors"
_NeurIPS.cc/2025/Conference — NeurIPS 2025 poster_

### Official Review · Reviewer_EHr8 · 2025-06-27

**Clarity:** 2
**Significance:** 3
**Originality:** 3
**Rating:** 5
**Confidence:** 2

**Summary:**

To have interpretable model, concept bottleneck models require expensive human expert annotation on the data. This paper aims to reduce this cost by letting LLMs generate the concept and annotate the dataset instead. The authors propose a split-sample update to reduce the impact of LLM hallucination. Experiment results show the proposed method performs better than interpretable baseline models.

**Questions:**

- I still don't fully understand how to use the proposed method to make prediction on a new data point. I think we need to first let the LLM annotate the new data point based on learnt concepts, but then how do you make prediction? Also, how do you account for the error that might occur during the LLM annotation?

- Citation is missing in Line 234 in appendix.

**Ethical Concerns:**

["NO or VERY MINOR ethics concerns only"]

**Final Justification:**

My concerns regarding the interpretability is successfully addressed by the rebuttal and therefore I raise my score to 5 (accept).

**Limitations:**

See weakness above.

**Quality:**

3

**Strengths And Weaknesses:**

Strength
- The problem this paper aims to solve is quite interesting.
- The idea of using LLMs within a rigorous Bayesian framework to achieve reliability is quite neat.

Weakness
- No thorough analysis is conducted on the LLM extracted concept. It is unclear how interpretable or trustworthy these concepts actually are.
- In the iterative procedure of generating new concepts, in each iteration, all training data are used. This can be highly inefficient once the data set gets larger and/or the iteration becomes large.
- Calibration is measured by Brier, which seems to me is an odd choice. ECE is a more standard measurement of calibration in the literature?
- The writing could be improved, I find the paper hard to follow. Also, the notation is a bit messy. A vector is represented by bold symbol and upper arrow at the same time.

---

> ### Author Rebuttal · Authors · 2025-07-30
>
> We thank the reviewer for their thoughtful comments and are heartened to hear that you find the method quite neat\! This work was inspired by our real-world collaboration with the local hospital to build risk prediction models, where there is a major need for methods that learn fully interpretable clinical risk prediction models for unstructured data. We believe BC-LLM, by bringing together the world knowledge of LLMs and the rigor of Bayesian inference, has the potential to address this critical gap. Thus we sincerely appreciate the reviewer's feedback on how the manuscript can be further improved and address the reviewer's points below:
>
> > No thorough analysis is conducted on the LLM extracted concept. It is unclear how interpretable or trustworthy these concepts actually are.
>
> Thank you for this question\! We can use the semi-synthetic MIMIC experiment to check how trustworthy the extracted concepts are, because we have the human annotations for the true concepts. In particular, we have human annotations for “Does the note mention the patient consuming alcohol in the present or the past?”, “Does the note mention the patient smoking in the present or the past?”, “Does the note mention the patient using recreational drugs in the present or the past?”, “Does the note imply the patient is unemployed or retired?”. BC-LLM converged to similar concepts with the following agreement rates with the human annotations:
>
> * For the alcohol concept, BC-LLM learned concept questions like “Does the note mention this patient having a history of alcohol use disorder?” and “Does the note indicate that the patient consumes alcohol?”. The agreement rate of LLM extractions with human annotations is 91.5%.
> * For the tobacco concept, BC-LLM learned concept questions like “Does the note mention the patient’s smoking history?” and “Does the note mention the patient as currently smoking?”. The agreement rate of LLM extractions with human annotations is 85.9%.
> * For the recreational drugs concept, BC-LLM learned concept questions like “Does the note mention the patient having a history of cocaine use?”. The agreement rate is 95.1%.
> * For the unemployment/retired concept, BC-LLM learned concept questions like "Does the note mention the patient being retired?" and "Does the note mention the patient as unemployed?". The agreement rate is 73%.
>
> So in this experiment, we found that the LLM extracted concepts are quite trustworthy. In manual reviews for other experiments, we also found the extracted concepts to be quite accurate. That said, in settings where the LLM cannot extract certain concepts accurately, we expect BC-LLM to converge towards concepts that it can accurately extract *and* are predictive of the label.
>
> > In the iterative procedure of generating new concepts, in each iteration, all training data are used. This can be highly inefficient once the data set gets larger and/or the iteration becomes large.
>
> Because the primary goal of this manuscript is to introduce a Bayesian formalization for learning CBMs, there is still room for improvement when it comes to the scalability of BC-LLM. Currently, the runtime BC-LLM is linear in the number of observations since it uses all the training data in each iteration. A straightforward way to speed this up is by using mini-batches instead. Our preliminary experiments show that BC-LLM achieves similar results when doing this for the greedy warm-up procedure. We are now speeding up the Gibbs sampling procedure with mini-batching while preserving statistical guarantees, using methods that estimate the MH acceptance ratio (see e.g. Seita 2018 and Bardenet 2014). We believe this is beyond the scope of this paper and thus leave it to future work.
>
> > Calibration is measured by Brier, which seems to me is an odd choice. ECE is a more standard measurement of calibration in the literature?
>
> Both Brier and ECE are standard measurements of calibration. We had selected Brier because it is also a proper scoring rule, so the best possible model will also attain the lowest Brier score. ECE tends to be very noisy for smaller dataset sizes since the ECE is highly sensitive to binning strategy; in addition, inaccurate models can also attain low ECE values.
>
> In the case of the semi-synthetic MIMIC experiment, the true probabilities are known, so the Brier score–defined here as the mean squared error between the true and predicted probabilities–is exactly what we want. As shown in the results, BC-LLM achieves a lower Brier score than other methods.
>
> For the experiments where the true probability is not known, all the methods have similar ECE values (using 5 bins). For CUB, we find that the BC-LLM attains an ECE of 0.25 with a 95% CI of (0.17, 0.32). This overlaps with the comparators LLM+CBM (ECE=0.23), Boosting LLM+CBM (ECE=0.24), Human+CBM (ECE=0.26), LLM+CBM (no keyphrases) (ECE=0.31), ResNet (ECE=0.26). For the readmission risk prediction experiment in Section 4.3, we find that BC-LLM attains an ECE of 0.184 with a 95% CI of (0.176, 0.193), which again overlaps with comparators LLM+CBM (ECE=0.182), Boosting LLM+CBM (ECE=0.180), and Bag-of-Words (ECE=0.177).
>
> > I still don't fully understand how to use the proposed method to make prediction on a new data point. I think we need to first let the LLM annotate the new data point based on learnt concepts, but then how do you make prediction? Also, how do you account for the error that might occur during the LLM annotation?
>
> Correct – To make a prediction on a new data point, the LLM annotates it with the learnt concepts. Each CBM in the ensemble (i.e., the posterior samples) then outputs a prediction per its fitted parameters. The final prediction is an average across the ensemble.
>
> BC-LLM does not additionally adjust for annotation error. Instead, it relies on the Bayesian inference procedure to implicitly do this adjustment. That is, concepts that the LLM cannot annotate accurately will have a low MH acceptance rate because these concepts will not be predictive of the target in held-out data. Instead, BC-LLM will converge towards concepts that both (i) the LLM is able to correctly annotate and (ii) are predictive of the target.
>
> > The writing could be improved.
>
> Thanks for this feedback. We will revise the notation to use only bold symbols for vectors, add a notation table for clarity, and clean up writing wherever possible to make the paper easier to follow.

---

> > ### Comment · Reviewer_EHr8 · 2025-08-01
> >
> > Thank the author for the response. It has addressed my concerns.

---

> > > ### Author Response · Authors · 2025-08-01
> > >
> > > We are glad we addressed your concerns! If possible, we respectfully ask you to consider raising your score. Thank you!

---

> > > > ### Comment · Reviewer_EHr8 · 2025-08-06
> > > >
> > > > I've increased my score to 5 and I hope the authors can deliver their promises in the final version.

---

> > > > > ### Author Response · Authors · 2025-08-06
> > > > >
> > > > > Thank you for engaging in this discussion and we truly appreciate you raising your score! We will definitely address your comments in the final version.

---

### Official Review · Reviewer_igan · 2025-07-02

**Clarity:** 3
**Significance:** 2
**Originality:** 2
**Rating:** 3
**Confidence:** 4

**Summary:**

The paper introduces a novel method that incorporates Large Language Models (LLMs) into Concept-Based Models (CBMs) using a Bayesian framework, with inference performed via Metropolis-Hastings sampling. The authors suggest that LLMs can act as informative priors, effectively guiding concept discovery, particularly in scenarios with limited prior information. Experiments on medical applications and bird image data suggest that  the proposed framework yields improved performance compared to the considered baselines.

**Questions:**

Even though the authors consider MC approaches, there are some other works in the literature concerning variational methods for concept selection. Wouldn't it also make sense to prompt the LLM once for concepts and then consider variable selection methods for the final subset?

I kindly ask the authors to provide the wall time measurements for the considered experiments.

Did the authors compare their approach to standard CBM benchmark settings such as CUB with the full 200 classes and not the 37 "families"? Did the authors also consider, SUN, Places365, ImageNet, or any other image dataset?

Can the authors elaborate on the OOD setup? Did the authors consider all the remaining 50% of the data (the testing split) for their computations? Can the authors also provide an accuracy metric for OOD?

In p.7 line 308 the authors mention "The difference when applying the Bunting bird CBM to an actual bunting bird versus an OOD sample". What is a Bunting bird CBM? The authors consider a single CBM with multiple prediction targets correct?

**Ethical Concerns:**

["NO or VERY MINOR ethics concerns only"]

**Final Justification:**

Taking into consideration the discussion with the authors and their responses to my and the other reviewers' concern, I raise my score from 2 to 3. Even though I find the use of a Bayesian approach appropriate and novel in the context that it is used, I still find that the manuscript is missing some important information/experiments/discussion to fully justify the contribution. These pertain to the comparison with other CBM approaches and the highly irregular experimental regime as noted in the initial review. Even though I can understand the benefit in the "low-knowledge" regime, I still found that there were some conflicting statements, e.g., using a more targeted LLM for those settings, that could potentially invalidate the contribution of the approach (given that a domain appropriate LLM could yield similar or better results to the proposed approach).

The authors did mention that some additional ablation studies and experiments will be included in the revised manuscript but I cannot assess the results without having them. Thus, I still believe that the approach is missing some key elements towards publication, that is why I cannot recommend acceptance at this time.

**Limitations:**

No, there isn't a dedicated limitations subsection.

Even if some potential issues of the proposed framework are hinted at various parts of the manuscript, it should be clear for the readers what are the limitations of this work.  For example, one limitation seems to be complexity. Even though the authors discuss the complexity in terms of O() notation and then mention that ", BC-LLM was quick to run, as one iteration takes less than a minute for text data and a few minutes for image data on a normal laptop". Another limitation can be the impact of the design of the input prompt or the parameters used for the LLM/VLM. None of these are discussed.

**Paper Formatting Concerns:**

No obvious issues with the formatting.

**Quality:**

2

**Strengths And Weaknesses:**

The proposed methodology—approaching concept refinement from a Bayesian perspective and leveraging LLMs as informative priors rather than mere knowledge sources—is both novel and well-motivated. This framing is particularly compelling given the challenge of applying Concept-Based Models (CBMs) in real-world settings where concept annotations are scarce.

The paper begins by listing well-known limitations of  CBMs—such as restricted context due to training data, computational cost, the trade-off between interpretability and accuracy and hallucinations. Even though these are indeed issues of CBMs,  I find that the authors do not convincingly show how the proposed method overcomes these issues. In terms of accuracy/interpretability, the authors base their experimental evaluation on a highly irregular regime that may be more favorable for the proposed framework but it does not provide adequate comparison to SoTA CBM approaches. At the same time, the interpretability evaluation of the approach in the context of augmenting a tabular model with clinical notes was only evaluated by "four clinicians", rendering the interpretation of the results ambiguous.

The computational cost is again something that is not clear. Yes, some CBM methods do use thousands of concepts but there are recent methods that consider sparsity inducing methods to effectively select the concepts; at the same time, I'm confident that using GPT4o-mini for querying for relevant concepts will be more costly (see also the limitations section).

In this context, the authors employ GPT-4o-mini—a vision-language model rather than a pure LLM—which introduces ambiguity regarding the method's dependence on prior visual/textual knowledge. The claim that BC-LLM outperforms other LLM-based CBMs in low-knowledge regimes is not convincingly demonstrated here (and leakage that was mentioned in the context of standard CBMs can also be an issue here).

The authors mention that " we prove that BC-LLM can provide rigorous statistical inference and uncertainty quantification" and this claim is spread throughout the manuscript; however, it is weakly supported by the experimental evaluation and it's only captured by the Brier metric and the entropy in the OOD setting but without substantial insights.

One other major concern lies in the design of the prompts used (appendix C), and since the quality and effectiveness of concept extraction can be highly sensitive to prompt phrasing, a more thorough analysis—such as a prompt ablation study—would substantially strengthen the experimental section.

Overall,  this work aligns with a growing trend of leveraging LLMs for enhancing interpretability, particularly within the context of Concept-Based Models (CBMs). While the authors aim to improve concept quality through a statistical framework, the fundamental limitations of using LLMs remain largely unaddressed. The method relies heavily on the LLM at multiple stages of the pipeline, making the resulting interpretations highly sensitive to factors such as the specific model used, decoding parameters (e.g., temperature), and the nature of the task itself. As a result, the robustness and consistency of the interpretations may vary significantly, raising concerns about their reliability and generalizability. At the same time, the experimental evaluation, in my view, does not demonstrate substantial improvement or provide the necessary insights to justify the original claims of the authors.

---

> ### Author Rebuttal · Authors · 2025-07-28
>
> We thank the reviewer for their detailed comments and for recognizing that our method is both novel and well-motivated. Before addressing each of the reviewer’s points, we believe it would be fruitful to highlight that the goals of this work significantly differ from prior work. The goal is to learn a fully-interpretable non-leaky CBM in the “low-knowledge regime” where it is difficult to list potentially relevant concepts upfront. This is a major unaddressed need in healthcare, as clinical providers often want fully interpretable, faithful models that can predict (often noisy) outcomes using the unstructured data and have limited insight into which candidate concepts are good to include. In fact, BC-LLM is inspired by our real-world collaboration with Anonymous hospital to build a risk prediction model, which is the experiment in Section 4.3 of the paper. As illustrated in this experiment, existing CBM methods do poorly because they are not designed for the low-knowledge regime. In contrast, BC-LLM does much better.
>
> **Choice of experiments**: The reason we do not copy existing experiments from other CBM works is that they are not designed to test the ability of a CBM to learn concepts in low-knowledge regimes. Existing works test the setting where the LLM already has good knowledge of the prediction task and is able to list a comprehensive list of candidate concepts. In contrast, our experiments are designed to simulate the low-knowledge regime as follows:
>
> * In the MIMIC example, we have real-world clinical notes and simulate the label Y based on a hand-annotated set of concepts. Because the label Y is simulated and we only tell the LLM the outcome is “some label Y”, the LLM is unlikely to know what set of concepts we actually used in the simulation. Furthermore, using this semi-synthetic setup, we can evaluate if BC-LLM converges to the known true concepts.
> * In the readmission risk prediction example based on real-world clinical notes, we use a dataset that is private to the Anonymous hospital, thereby eliminating the possibility of an LLM having seen the data. Furthermore, readmission is known to be very difficult to predict, and factors influencing readmission are known to be highly variable across different hospitals. The factors picked up by BC-LLM are indeed *uniquely* relevant to Anonymous hospital, where a substantial subgroup of patients exhibit substance use.
> * For the CUB example, we designed the task to be slightly different from prior setups to minimize the chance of the LLM already knowing the answer. To do this, we do not reveal to the LLM that the task is to predict bird species and change the prediction task. Rather than predicting one of 200 bird species, the task is to predict a bird species within its bird family (a set of closely related bird species, e.g. a Bunting bird family is composed of three different Bunting bird species). Furthermore, predicting within bird families is better aligned with the fact that (i) our primary goal is to learn highly interpretable CBMs with fewer concepts and (ii) there likely exists a smaller CBM for distinguishing bird species within a family. In contrast, a CBM with a large number of concepts is no longer easily interpretable.
>
> Additional experiments, including OOD performance: We have added the following
>
> * *Original CUB-200 dataset*
> * *Functional Map of the World* (fMoW) (Christie et al., 2018): RGB satellite images classified into 62 building/land use categories. Models were trained on satellite images from the US, with 100 images sampled from each category. CBMs were trained to have 60 concepts. Models were evaluated in terms of in-distribution performance (i.e. images from the USA) as well as out-of-distribution performance when evaluated on satellite images from China. As another reviewer mentioned, these satellite images may require more domain expertise than the CUB dataset included in the original submission.
> * *Imagenette*: 10 classes from ImageNet. CBMs were trained to have 10 concepts.
>
> The table below reports the accuracy of BC-LLM as well as comparators. As we see, BC-LLM outperforms the CBM methods and even outperforms ResNet on the fMoW dataset.
>
> |  | fMoW (USA images) | fMoW (OOD-China images) | Imagenette | CUB-200 (prediction for all 200 bird species) |
> | :---- | :---- | :---- | :---- | :---- |
> | BC-LLM | **0.36** | **0.27** | **0.99** | 0.56 |
> | LLM+CBM | 0.31 | 0.22 | 0.90 | 0.47 |
> | Boosting LLM+CBM | 0.12 | 0.08 | 0.64 | 0.005 |
> | ResNet | 0.30 | 0.17 | **0.99** | **0.74** |
>
> **Adequate comparison to SoTA CBM approaches**: To our knowledge, the paper includes all major non-leaky CBM comparators: (A) Fitting a CBM on human-selected and annotated concepts (referred to as “Human+CBM”), (B) Using an LLM to suggest and annotate concepts and then fitting a variable selection method on these concepts (referred to as “LLM+CBM (no keyphrases)”), (C) Using an LLM to brainstorm keyphrases that describe each observation, fitting a sparse model on these keyphrases, and having the LLM suggest concepts based on the top keyphrases (“LLM+CBM”), and (D) Using an LLM to suggest concepts in a boosting procedure (“Boosting LLM \+ CBM”). Note that we do not include CBM methods that rely on “soft” (i.e. continuous-valued) concept extractors, as they can suffer from information leakage and thus are not fully interpretable.
>
> **Interpretability evaluation**: We appreciate these concerns and will ask for input from more clinicians for the revised manuscript. That said, we would like to highlight that the average relevance score for BC-LLM is \>1 standard error higher than that for the other methods. Getting input from clinicians is rare for AI/ML conference papers, and having input from four clinicians is substantially more than most. We will include error bars to Fig 4 in the revised manuscript.
>
> **Rigorous statistical inference and uncertainty quantification**: BC-LLM is supported by rigorous theoretical analyses, faster convergence to the correct concepts, lower Brier score, and higher entropy in OOD settings. We have also demonstrated better performance in OOD settings in the additional experimental results shown above.
>
> **Computational cost**: The average wall times for BC-LLM are: 1 minute per epoch for Section 4.1 (short clinical notes), 3 minutes per epoch Sec 4.2 (CUB), and 1.2 minutes per epoch for Section 4.3 (long clinical notes). Each BC-LLM experiment was run for 5 epochs. The cost for BC-LLM is around \\$0.0015 per request for CUB. On average, this translates to around \\$3 per experiment. The cost for MIMIC is around \\$0.0005 per request, which translates to around \\$1 per experiment. We believe these costs are highly reasonable. We note that this is all without requiring human annotations for the concepts, which is significantly more costly. Furthermore, the concepts in BC-LLM are not extracted through embedding similarity-based measures, which are known to be leaky and more error-prone.
>
> **Ablation studies to study sensitivity to LLM dependence**: We have investigated sensitivity to the choice of the LLM in Section D of our appendix, where we show results from using BC-LLM with Cohere’s Command-R and Claude 3.5 Haiku instead. We consistently find that BC-LLM outperforms comparator methods. We used a temperature of zero to minimize stochasticity and default parameters otherwise. We used very basic prompts (Section C of our appendix) to reflect the “low-knowledge” regime. The revised manuscript will explore how performance changes if we conduct further prompt engineering.
>
> **Related works**: We have additionally added some discussion on variational methods such as [2]
>
> We hope our responses clarify any points of confusion and demonstrate the novelty and importance of this work. If possible, we kindly ask the reviewer to reassess their score.
>
> \[1\] Ragkousis et al. "Tree-Based Leakage Inspection and Control in Concept Bottleneck Models" arXiv:2410.06352
>
> \[2 \] Panousis et al. "Coarse-to-Fine Concept Bottleneck Models" NeurIPS 2024

---

> > ### Comment · Reviewer_igan · 2025-08-06
> >
> > I thank the authors for their thorough responses to my concerns. The provided clarifications and additional experiments address most of my concerns. That being said, there are still some points that require further investigation. On the one hand, I can see the usefulness of the proposed approach in the low-knowledge regimes as the authors note, but on the other, the highly different experimental settings and the responses of the authors to other reviewers' concerns that potentially domain specific LLMs might yield better results cast some doubt on the resulting contribution and applicability of the proposed work. Nevertheless, taking into consideration the effort of the authors and their responses to all the concerns, I will increase my score and will further evaluate the provided information for the next phase.

---

> > > ### Author Response · Authors · 2025-08-06
> > >
> > > Thank you so much for this discussion and we appreciate the detailed suggestions on how to improve the work. We will certainly revise the manuscript to address your comments.
> > >
> > > We sincerely thank the reviewer for increasing their score. We kindly ask you to double check that the score in OpenReview has been updated accordingly. While we do not see the final score, it seems that scores disappear the moment a reviewer modifies their score. Thanks!

---

### Official Review · Reviewer_EwWN · 2025-07-03

**Clarity:** 3
**Significance:** 3
**Originality:** 2
**Rating:** 4
**Confidence:** 3

**Summary:**

This paper proposes BC-LLM, a Bayesian framework that enhances Concept Bottleneck Models (CBMs) by leveraging Large Language Models (LLMs) to explore an open-ended space of interpretable concepts. Unlike standard CBMs that rely on a fixed concept set, BC-LLM uses LLMs as both concept extractors and priors, enabling scalable and flexible concept discovery with uncertainty quantification. Despite potential LLM hallucinations, the authors provide theoretical guarantees. Experiments across image, text, and tabular domains show that BC-LLM outperforms interpretable and black-box baselines, converges faster to relevant concepts, and is more robust to distribution shifts.

**Questions:**

1. Generalizability to Other Image Domains:
Can this approach be applied to different image classification datasets beyond those used in the paper?


2. Applicability to Domain-Specific Tasks:
How would this method perform on tasks that require highly domain-specific knowledge, where LLMs may lack sufficient prior understanding, such as satellite imagery or industrial/factory imaging? Is BC-LLM still applicable or effective in these cases?

3. Prompt Design for Concept Annotation:
Could you provide the exact prompt used in Step 3 for concept annotation in the bird image classification task (Appendix)? How are concept annotations generated across different modalities?


4. Entropy in OOD Settings:
In Table 1, the ResNet model shows the highest entropy in the out-of-distribution (OOD) setting. Could you clarify why this might be the case?

5. Related Work Suggestion:
You may want to cite the paper "Constructing Concept-based Models to Mitigate Spurious Correlations with Minimal Human Effort" (ECCV 2024), which also uses LLMs for concept candidate generation, filtering, and annotation. It seems closely related and could help contextualize your contribution.

**Ethical Concerns:**

["NO or VERY MINOR ethics concerns only"]

**Final Justification:**

The reviewer addressed some of my concerns during the rebuttal. However, I still find that the approach is limited in domains that require highly specific knowledge not captured by LLMs. Therefore, I will maintain my original score.

**Limitations:**

yes

**Paper Formatting Concerns:**

no major issues

**Quality:**

3

**Strengths And Weaknesses:**

Strengths:

1. Presents a novel approach to building Concept Bottleneck Models (CBMs) by leveraging Large Language Models within a Bayesian framework.


2. Demonstrates versatility by applying the method across multiple modalities, including image, text, and tabular data.


3. Offers a more lightweight yet effective CBM by selectively identifying and using only the most relevant concepts, reducing annotation cost and complexity.

Weaknesses:

1. Although the method is evaluated across different modalities, the dataset diversity—particularly for complex, real-world image domains, is limited. This raises concerns about the method's robustness in settings where human-defined concepts are ambiguous or less clearly represented.

---

> ### Author Rebuttal · Authors · 2025-07-30
>
> We thank the reviewer for their thoughtful comments and for recognizing the novelty of this Bayesian framework for learning CBMs. This work was inspired by our real-world collaboration with the local hospital to build risk prediction models, where there is a major need for methods that learn fully interpretable clinical risk prediction models for unstructured data. We believe BC-LLM, by bringing together the world knowledge of LLMs and the rigor of Bayesian inference, has the potential to address this critical gap. Thus we sincerely appreciate the reviewer's feedback on how the manuscript can be further improved and address the reviewer's points below:
>
> > Generalizability to Other Image Domains and Applicability to Domain-Specific Tasks: Can this approach be applied to different image classification datasets beyond those used in the paper? How would this method perform on tasks that require highly domain-specific knowledge, where LLMs may lack sufficient prior understanding, such as satellite imagery or industrial/factory imaging?
>
> Thank you for these two questions about the robustness of BC-LLM to other real-world image datasets\! To illustrate the generalizability of BC-LLM to other image domains, we have run experiments on two additional datasets:
>
> * *Functional Map of the World* (fMoW) (Christie et al., 2018): RGB satellite images classified into 62 building/land use categories. Models were trained on satellite images from the US, with 100 images sampled from each category. CBMs were trained to have 60 concepts. Models were evaluated in terms of in-distribution performance (i.e. images from the USA) as well as out-of-distribution performance when evaluated on satellite images from China. As the reviewer mentioned, these satellite images may require more domain expertise than the CUB dataset included in the original submission.
> * *Imagenette* (https://github.com/fastai/imagenette): 10 classes from ImageNet. CBMs were trained to have 10 concepts. Note that we do not run on all of ImageNet, since BC-LLM is not intended for learning CBMs with a very large number of classes.
>
> The table below reports the accuracy of BC-LLM as well as comparators (we exclude human+CBM because there are no available human-generated concepts for these datasets). As we see, BC-LLM outperforms the CBM methods and even outperforms ResNet on the fMoW dataset.
>
> |  | fMoW (USA images) | fMoW (OOD-China images) | Imagenette |
> | :---- | :---- | :---- | :---- |
> | BC-LLM | **0.36** | **0.27** | **0.99** |
> | LLM+CBM | 0.31 | 0.22 | 0.90 |
> | Boosting LLM+CBM | 0.12 | 0.08 | 0.64 |
> | ResNet | 0.30 | 0.17 | **0.99** |
>
> While these results are promising, we acknowledge that for highly specific problem domains, BC-LLM may require some tailoring. For instance, rather than using a general LLM as we do in the experiments, one may instead need to use a domain-specific LLM (e.g. a medical LLM). Furthermore, if there is no LLM that can accurately extract values for domain-specific concepts in a zero-shot manner, one may need to train/tune a model specifically to extract domain-specific concepts.
>
> > Prompt Design for Concept Annotation: Could you provide the exact prompt used in Step 3 for concept annotation in the bird image classification task (Appendix)? How are concept annotations generated across different modalities?
>
> Here is the prompt used for concept annotation (Step 3\) in the bird image classification task. The image is uploaded to the LLM through the API alongside the prompt
>
> *"You will be given an image. I will give you a series of questions. Your task is to answer each question with 0 or 1\. You may output a probability if you are unsure, but avoid this in general. Summarize the response with a JSON that includes your answer to all of the questions. Questions:*
>
> 1. *\<CONCEPT QUESTION 1\>*
>
> 2. *\<CONCEPT QUESTION 2\>*
>
> 3. *…*
>
> *Output your answer in JSON"*
>
> We fill in the prompt with the concept questions for which we want annotations.
>
> > Entropy in OOD Settings: In Table 1, the ResNet model shows the highest entropy in the out-of-distribution (OOD) setting. Could you clarify why this might be the case?
>
> ResNet can achieve higher entropy than CBM models because CBMs are highly constrained: CBMs have only a small number of input features by design, and the values of these input features are further constrained to be interpretable (e.g., a probability). As a consequence, a CBM may not be able to fully represent the additional uncertainty in OOD settings. Nevertheless, because a CBM is easy to audit by humans, it is also easier to debug than a ResNet model. By simply observing the extracted concept values, one is likely able to detect whether an observation is OOD. Furthermore, as shown in the new experiment with fMoW, we find that BC-LLM can outperform ResNet in terms of OOD performance in certain cases.
>
> > Related Work Suggestion: You may want to cite the paper "Constructing Concept-based Models to Mitigate Spurious Correlations with Minimal Human Effort" (ECCV 2024), which also uses LLMs for concept candidate generation, filtering, and annotation. It seems closely related and could help contextualize your contribution.
>
> Thank you for this suggestion – we have added a discussion on it to our related works section. The paper highlights studying correlated concepts and is very similar to the “LLM+CBM” baseline we compare against. The iterative Bayesian framing we propose provides an extra layer of uncertainty quantification and enables finding more concise models than the method proposed in the suggested paper.

---

> > ### Comment · Reviewer_EwWN · 2025-08-06
> >
> > Thank you for the clarification and for providing additional experimental results. While some of my concerns were addressed in the response, I still find that the approach is limited in domains that require highly specific knowledge not captured by LLMs. Therefore, I will maintain my original score.

---

### Official Review · Reviewer_JKHs · 2025-07-03

**Clarity:** 3
**Significance:** 3
**Originality:** 2
**Rating:** 5
**Confidence:** 4

**Summary:**

This paper presents a BC LLM, i.e., a bayesian approach for concept bottleneck models. The key insight here is to use LLMs as priors and concept annotators while doing inference via a split-sample metropolis-within-Gibbs scheme. The paper also presents a theorem that suggests that the markov chain defined by running this split-sample scheme then the BC-LLM will asymptotically recover the true set of concepts that optimize the log-likelihood.  The BC-LLM is initialized with top concepts from an LLM. The concepts are then refined via a query replacement scheme. The paper then shows the benefits of the BC-LLM for bird-classification, clinicals, and for a tabular model.

**Questions:**

- How can one recover from poor-quality concept suggests at the start?
- What is the cost for additional K? Specifically, how do you expect the computational cost to change with increasing concept size.
- Can the LLM hallucinate concepts valid for training, but not relevant for test? How should the uncertainty adapt in this case?

**Ethical Concerns:**

["NO or VERY MINOR ethics concerns only"]

**Final Justification:**

I have read the rebuttal from the authors, and their response. I believe this is a worthy contribution. They have successfully addressed my concerns regarding initial concept suggestion quality, and scale.

**Limitations:**

- Scalability to larger concepts as the paper mentioned.

**Paper Formatting Concerns:**

None.

**Quality:**

4

**Strengths And Weaknesses:**

## Strengths

- Interesting Theorem to justify selection: the theorem offers a consistency guarantee, BC-LLM will asymptotically recover the true set of concepts that optimize the log-likelihood.

- The result is inspired by a well-established literature on Bayesian model selection and is nontrivial to extend to an infinite concept space proposed by LLMs. The split-sample scheme is nice, and practically effective.

- Evaluation is thorough, diverse, and realistic: covering simulated and real-world clinical text, vision (CUB birds), and multi-modal EHR data.

## Weaknesses
- Unclear implication of assumption H.1. It would be great to include a high-level summary of the justification for this.
- What if all the initial concepts suggested by the LLM are not related to the task? As with all metropolis hastings like schemes, the sampling complexity here might be high.
- Scalability, to large number of concepts, is a challenge, but the authors already mention this.
- Some of the figures are small and could be bigger.

- Ablation against LLM+CBM and Boosting LLM+CBM are fair and convincing. Concept interpretability was validated through expert clinician evaluation, adding real-world credibility.

---

> ### Author Rebuttal · Authors · 2025-07-30
>
> We thank the reviewer for the thoughtful comments. We are heartened to hear that you found the Bayesian framework to be interesting and nontrivial and the experiments to be convincing. This work was inspired by our real-world collaboration with the local hospital to build risk prediction models, where there is a major need for methods that learn fully interpretable clinical risk prediction models for unstructured data. We believe BC-LLM, by bringing together the world knowledge of LLMs and the rigor of Bayesian inference, has the potential to address this critical gap. Thus we sincerely appreciate the reviewer's feedback on how the manuscript can be further improved and address the reviewer's points below:
>
> > Unclear implication of assumption H.1. It would be great to include a high-level summary of the justification for this.
>
> Assumption G.1. is used to prove Theorem 3.1, which states that MH sampling with a potentially misspecified LLM prior and proposal distributions can still be robust and converge to a desired target distribution. This result is highly unique, because most works on Bayesian analysis analyze an explicitly defined posterior and do not consider the effect of misspecification in the sampling process on the target distribution. Thus additional assumptions are required. Most components in Assumption G.1 are either standard regularity conditions ((i), (ii), (iv), (v), (vi), (viii), (ix)) that are often made in statistical analysis or are hyperparameter choices ((iii) and (x)). The key new assumption is (vii), which asserts that at each step, there is at least a fixed nonzero probability of the LLM proposing one of the K “best” concepts conditional on the data. This helps to ensure at every outer loop of the sampler (i.e. for every cycle through all the positions of the concept vector), enough mass flows to the optimal set of concepts. We believe this is a reasonable assumption even in regimes where the LLM has limited knowledge, as long as the degree of knowledge is non-zero.
>
> > What if all the initial concepts suggested by the LLM are not related to the task? As with all metropolis hastings like schemes, the sampling complexity here might be high.
>
> Thank you for this question! First, we would like to highlight that this work was motivated by this very concern you have raised. Most existing methods ask LLMs for a set of candidate concepts or rely on a pre-existing concept list, but this is not effective when LLMs or humans have limited prior knowledge of which concepts are truly relevant. BC-LLM addresses this by iteratively refining the set of concepts during posterior inference.
>
> Second, we agree that if an MH sampling procedure is initialized poorly, sampling complexity will be high. Because of this very concern, the initialization and warm-up steps of BC-LLM are specifically designed to accelerate convergence. For initialization, the LLM explores a large variety of concepts by brainstorming keyphrases that describe each observation and then selects concepts based on the keyphrases that are most predictive. For warm-up, the LLM greedily chooses concepts that improve prediction performance, again based on keyphrases that are most associated with the residual. Together, these two steps ensure the MH sampling procedure starts closer to the posterior.
>
> > Scalability to a large number of concepts (K) is a challenge… How do you expect the computational cost to change with increasing concept size?
>
> Because the primary goal of this manuscript is to introduce a Bayesian formalization for learning CBMs, there is still room for improvement when it comes to the scalability of BC-LLM. Currently, the runtime of BC-LLM is linear in the number of concepts K, because each iteration of Gibbs refines/replaces a single concept. A straightforward way to speed this up is by refining J>1 concepts in each iteration, which reduces the runtime by a factor of J. Our preliminary experiments show that BC-LLM achieves similar results when doing this for the greedy warm-up procedure. However, extending the Gibbs sampling procedure to refine/replace multiple concepts in a single iteration while preserving statistical guarantees is beyond the scope of this paper, and we leave it to future work.
>
> > Can the LLM hallucinate concepts valid for training, but not relevant for test? How should the uncertainty adapt in this case?
>
> During the posterior sampling procedure, it is certainly possible for the LLM to propose irrelevant concepts (e.g., because it fits the training data well). Nevertheless, for a sufficiently large dataset, irrelevant concepts are likely to be rejected since they don’t fit the held-out data well. For small datasets where there is insufficient sample size to distinguish between relevant and irrelevant concepts, irrelevant concepts are more likely to be accepted, but the posterior sample will also reflect this increased uncertainty by including a larger variety of concepts. Indeed, we observe such behavior in Figure 3 (right): the posterior samples for the smaller dataset reflect high uncertainty (many unique concepts) while the posterior samples for the larger dataset reflect low uncertainty (few unique concepts).

---

> > ### Comment · Reviewer_JKHs · 2025-08-05
> > **Satisfactory response**
> >
> > Hi,
> >
> > Thanks for the feedback and response. I think this is compelling work that provides a different take on how to incorporate LLM sampled concepts. Ultimately, as we rely on LLMs more for concept list suggestion and use, schemes like the one proposed here will be useful especially in critical settings. I read the other reviews, and think the additional experiments justify me maintaining my score.

---

### Note · Authors · 2025-08-12

We thank the AC and reviewers for their time, engagement, and thoughtful feedback!

We are glad to see the generally positive reception by the reviewers, including their appreciation of the problem we address, e.g. calling it “quite interesting” (EHr8) or “novel and well motivated” (igan) and appreciation for the method we propose, e.g. calling it “nontrivial” (JKHs), “lightweight yet effective” (EwWN), and “thorough, diverse, and realistic” in evaluation (JKHs). We found a few misunderstandings in the reviews that we have addressed in our author rebuttals.

Given the extensive discussions on the details of the method, we’d like to consider the broader context and overarching goal of this work. In high-stakes settings such as healthcare, one of the major unresolved problems is learning fully interpretable prediction models for unstructured data, especially when it is difficult to list relevant concepts upfront (from a human or LLM). We have encountered this repeatedly in our own collaborations with clinical care providers. While CBM methods have made some progress, they continue to struggle in this area due to issues such as information leakage.

To address this methodological gap, the novel approach in this work is to iteratively refine CBMs through Bayesian posterior sampling, which allows us to search directly over the space of concepts in a statistically rigorous manner. This is critical, as it allows us to ensure the CBM is fully interpretable. Furthermore, this approach lets us quantify uncertainty over concepts, integrate an LLM prior when available, and correct for poor concept proposals. Empirical results show that the learned models attain strong predictive performance across diverse datasets. Theoretical results show that the method will find the relevant concepts, asymptotically, despite LLM outputs being unpredictable or their priors being inconsistent.

Again, we would like to thank the reviewers for their suggestions. We sincerely believe these suggestions will further improve the work and will make sure to incorporate them into the revised version.

---

### Decision · Program_Chairs · 2025-09-17

**Decision:**

Accept (poster)

**Comment:**

This paper addresses the problem of model interpretability through human-interpretable concepts via secondary models (like concept bottleneck models, CBMs). Unlike most CBMs, which trade-off interpretability and accuracy, this paper proposes to use LLMs as priors within a Bayesian formulation to sample and refine concepts through posterior sampling ideas. The authors show that their approach asymptotically recovers the correct concepts, provides notions of uncertainty quantification, comprises interesting empirical results comparing with standard CMBs, and features clinician validation in a clinical example.

**Strengths**
* The Bayesian framework for interpretation with the prior information contained in an LLM is a new and interesting perspective to the interpretability problem, agreed upon by all reviewers.
* Their split-sample scheme is a simple yet effective strategy to provide guarantees.
* The approach is applicable in a variety of modalities (image, text, and tabular data).
* Validation by clinicians on clinical example adds credibility.

**Weaknesses**
* Some assumptions were not clearly flashed out.
* Reviewers questioned aspects of computational costs of the approach, and their comparisons to other CBM methods.
* The choice of datasets lacked some diversity (they focused on "low-knowledge" regimes) and was not very standard (as opposed to "high SNR" regimes like Place365, etc).

**Discussion summary and conclusion**

The discussions between authors and reviewers was productive. Most weaknesses were addressed and resolved (e.g. by clarifying assumptions, providing further experimental results on new datasets and with further clinical evaluation, and further details on computational cost). Two reviewers provide strong support (5), one borderline accept (4) and one borderline reject (3). With the understanding that the authors will include the new results and promised details in their revised version of the manuscript, and based on the novelty of their proposed Bayesian approach, I recommend acceptance.